# Design of Ligand-Binding Proteins with Atomic Flow Matching

## Abstract

Designing novel proteins that bind to small molecules is a long-standing challenge in computational biology, with applications in developing catalysts, biosensors, and more. Current computational methods rely on the assumption that the binding pose of the target molecule is known, which is not always feasible, as conformations of novel targets are often unknown and tend to change upon binding. In this work, we formulate proteins and molecules as unified biotokens, and present ATOMFLOW, a novel deep generative model under the flow-matching framework for the design of ligand-binding proteins from the 2D target molecular graph alone. Operating on the positions of biotokens, ATOMFLOW captures the flexibility of ligands and generates ligand conformations and protein backbone structures iteratively. We consider the multi-scale nature of biotokens and demonstrate that ATOMFLOW can be effectively trained on a subset of structures from the Protein Data Bank, by matching the flow vector field using an SE(3) equivariant structure prediction network. Experimental results demonstrate that our method generates high-fidelity ligand-binding proteins, matching or surpassing the performance of RFDiffusionAA across multiple metrics—without requiring bound ligand structures. As a general framework, ATOMFLOW can be readily extended to diverse biomolecule design tasks in the future.

## 1 Introduction

Proteins are indispensable macromolecules that drive the essential processes of living organisms. A crucial mechanism by which they accomplish this is through binding with small molecules (Schreier et al., 2009). Continuous progress has been made to design ligand-binding proteins with various biological functions, such as catalysts and biosensors (Bennett et al., 2023). However, the problem remains challenging due to the complex interactions between proteins and molecules, as well as the inherent flexibility of ligands. The most well-established approaches depend on shape complementarity to dock molecules onto native protein scaffold structures (Bick et al., 2017; Polizzi & DeGrado, 2020), which are computationally expensive.

RFDiffusionAA (Krishna et al., 2024) is currently the leading model for de novo protein design targeting small molecule ligands. Based on the all-atom structure prediction model RoseTTAFoldAA (Krishna et al., 2024), it achieves strong performance in generating ligand-binding proteins and improves upon its predecessor RFDiffusion (Watson et al., 2023), which does not directly model protein-ligand interaction. A key limitation of RFDiffusionAA is its assumption that the ligand adopts a known and rigid binding conformation. This assumption does not hold for many ligands, especially

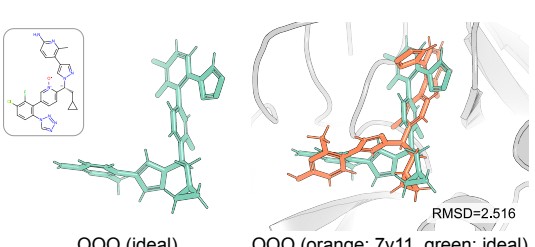

Figure 1: The conformer of OQO deforms upon binding to coagulation factor XIa. Green: ideal conformer. Orange: bound conformer.

those without known binding poses (Bick et al., 2017). Although diverse conformers can be sampled and filtered using expert heuristics (Krishna et al., 2024), this process is computationally intensive. In addition, ligands often exhibit conformational flexibility upon binding (Mobley & Dill, 2009), as shown in Figure 1. Recently, AlphaFold 3-like models have shown the ability to capture ligand flexibility during docking (Abramson et al., 2024). While they can be repurposed for binder design (Yang

et al.), achieving practical performance has so far required large-scale post-training, and existing work remains limited to CDR loop design and constrained by the fixed model architecture.

| | ATOMFLOW | RFDiffusionAA | RFDiffusion | AlphaFold 3 |
|---|---|---|---|---|
| *De novo* Design Capability | ✓ | ✓ | ✓ | ✓with repurposing |
| Contact-Based Modeling | ✓ | ✓ | ✗ | ✓ |
| Ligand Flexibility | ✓ | ✗ | ✗ | ✓ |
| Pretrained Model Independence | ✓ | ✗ | ✗ | ✗ |

Table 1: Comparison of key features across the methods.

To address the aforementioned issues, we present Atomic Flow-matching (ATOMFLOW), a novel deep generative model with a flow-matching framework (Lipman et al., 2022; Liu et al., 2022) on atomic biotokens for the design of ligand-binding proteins from 2D molecular graphs alone. Key features of ATOMFLOW are compared in Table 1. We model different types of biomolecules within a unified framework that operates in a shared spatial representation, which maximizes information aggregation (Bryant et al., 2024), with a flow matching model that directly designs the interactions. Instead of relying on a fixed ligand conformer, ATOMFLOW learns to update the ligand structure along with the structure of the protein binder. We define a flow on the representative atoms of the tokens as a linear interpolation between the bound protein-ligand complex structures and noisy structures and demonstrate that, with minor approximations, the vector field of the defined flow can be effectively learned using an SE(3)-equivariant structure module and a variant of Frame Aligned Point Error (FAPE) loss (Jumper et al., 2021) that compensates for the multi-scale nature of their geometric features.[1] The concept of regressing the vector field through structure denoising has also been explored in Jing et al. (2024), though their work focuses on conformer prediction and depends on a pre-trained structure prediction model.

We designed an *in silico* evaluation pipeline for this task, evaluating ATOMFLOW on multiple metrics following previous works (Krishna et al., 2024). We also introduce an alternative binding affinity metric based on the confidence scores of AlphaFold 3-like models (Abramson et al., 2024; team et al., 2024), which is experimentally validated to correlate strongly with binding potential in minibinder design (Zambaldi et al., 2024). The performance of ATOMFLOW outperforms or is comparable to RFDiffusionAA, with flexible ligand conformer modeling and more than 5x faster inference speed. A case study further highlights that when the bound structure is unknown, ATOMFLOW successfully designs protein binders with more interatomic contacts, whereas RFDiffusionAA can be constrained by its dependence on a fixed, suboptimal ligand structure.

## 2 RELATED WORK

**Ligand-binding Protein Design.** Traditional approaches to ligand-binding protein design mainly rely on docking molecules onto large sets of shape-complementary protein pockets (Polizzi & DeGrado, 2020; Lu et al., 2024). While the screening process can be accelerated with deep learning models (An et al., 2023), conventional methods are computationally expensive and often depend on domain experts (Bick et al., 2017). Recent advances in deep generative models have paved the way for data-driven approaches, and a variety of models have been proposed to design proteins conditioned on binding targets (Shi et al., 2022; Kong et al., 2023; Watson et al., 2023; Zhang et al., 2024). Focusing on molecule binder design, RFDiffusion (Watson et al., 2023) generates novel proteins from scratch, using a heuristic attractive-repulsive potential to measure shape complementarity. The follow-up work RFDiffusionAA (Krishna et al., 2024) improves the performance by explicitly modeling the interactions between proteins and molecules with an all-atom formulation. These approaches assume binding poses of ligands are known and impose rigidity constraints on ligand structures. Another line of research focuses on designing binding pockets for small molecules (Stark et al., 2024; Zhang et al., 2024). While taking ligand flexibility into consideration, they can only design the portions of proteins that interact with the ligands and require the rest part of the proteins as input. Our model also accounts for the ligand flexibility, but is able to design full ligand-binding proteins from 2D molecular graph alone.

**Protein Generative Model and Structure Prediction.** Recently, various deep generative models for protein generation have emerged (Ingraham et al., 2023; Lin & AlQuraishi, 2023; Yim et al.,

---

[1]The size of a protein is often much larger than that of a molecule. The size disparity should be considered when designing flow-matching models for stable training and inference.

2023b;a; Wu et al., 2024; Watson et al., 2023; Krishna et al., 2024). For example, Genie (Lin & AlQuraishi, 2023) introduces a diffusion process defined on C$\alpha$ coordinates of proteins and allows for the incorporation of motif structures as conditions. FrameDiff (Yim et al., 2023b) takes a step further by generating novel protein backbone structures using an SE(3) diffusion process applied to residue frames. Its successor, FrameFlow (Yim et al., 2023a), accelerates the generation process by leveraging the flow-matching framework. However, these approaches are tailored for single-chain protein generation and fall short in modeling multiple biomolecules. Recent development of all-atom strucutre prediction models such as RoseTTAFoldAA (Krishna et al., 2024) and AlphaFold 3 (Abramson et al., 2024) tokenize various types of biomolecules into unified tokens, aiming to develop a universal structure prediction model for all molecular types presented in the Protein Data Bank. Emerging work has explored repurposing structure prediction models like AlphaFold 3 as generators, though current efforts are limited to antibody CDR design rather than complete protein generation (Yang et al.). RFDiffusion 2 generates protein binders based on fixed fragmented pocket residues without the need to specify their sequential locations (Ahern et al., 2025). Our method adopts a similar formulation, leveraging unified tokenization to enhance information exchange between proteins and other biomolecules (Bryant et al., 2024).

## 3 PRELIMINARIES

### 3.1 NOTATIONS AND PROBLEM FORMULATION

**Notations.** In this work, a protein-ligand complex is represented as a series of $N$ biotokens $\{a_i \mid a_i = (s_i, x_i), i = 1, 2, \ldots, N\}$, where each token $a_i$ corresponds to either a protein residue or a ligand atom, $s_i$ denotes the token type, and $x_i \in \mathbb{R}^3$ denotes the token position, i.e. the coordinate of its representative atom. Let $\mathcal{S}_{\text{protein}}$ and $\mathcal{S}_{\text{atom}}$ be the set of amino acid types and chemical elements, respectively. For protein residues, $s_i \in \mathcal{S}_{\text{protein}}$, with $x_i$ being the position of the C-$\alpha$ carbon. For ligand atoms, $s_i \in \mathcal{S}_{\text{atom}}$, with $x_i$ being the atomic position. We define the protein token set as $\mathcal{P} = \{a_i \mid s_i \in \mathcal{S}_{\text{protein}}\}$, with $N_p = |\mathcal{P}|$ being the number of protein residues, and the ligand token set as $\mathcal{M} = \{a_i \mid s_i \in \mathcal{S}_{\text{atom}}\}$, with $N_m = |\mathcal{M}|$ representing the number of ligand atoms. In our settings, $N = N_p + N_m$. The biotokens are attributed with token-level features $f^{\text{token}} \in \mathbb{R}^{N \times c_t}$ and pair-level features $f^{\text{pair}} \in \mathbb{R}^{N \times N \times c_p}$, where $c_t$ and $c_p$ denote the feature dimensions.

**Problem Formulation.** Given a ligand molecule represented as a chemical graph $\mathcal{G} = (\mathcal{V}, \mathcal{E})$ and a residue count $N_p$ for the protein binder to be designed, we aim to generate a protein-ligand complex, where a conformer of $\mathcal{G}$ is docked to a protein binder with $N_p$ residues. Specifically, by describing the target protein-ligand complex as a series of biotokens, we generate the token positions $\{x_i\}$, with $\mathbf{x}_m = \{x_i \mid a_i \in \mathcal{M}\}$ being a valid conformer for $\mathcal{G}$, and $\mathbf{x}_p = \{x_i \mid a_i \in \mathcal{P}\}$ being a protein binder with high binding affinity to $\mathbf{x}_m$. Following previous works (Krishna et al., 2024; Yim et al., 2023b), we additionally generate the token frames $\{T_i = (r_i, t_i) \mid a_i \in \mathcal{P}\}$ for protein tokens as described in Appendix A.1, which can be used to recover full backbone coordinates of residues. The design of residue types $\{s_i \mid a_i \in \mathcal{P}\}$ is delegated to an existing reverse folding model (Dauparas et al., 2023).

### 3.2 FLOW MATCHING

Building upon the significant success of diffusion models in various generative tasks, flow matching models (Albergo & Vanden-Eijnden, 2022; Liu et al., 2022) allow for faster and more reliable sampling from a distribution learned from data. The generative process of flow matching models is usually defined by a probability path $p_t(x), t \in [0, 1]$ that gradually transforms from a known noisy distribution $p_0(x) = q(x)$, such as $\mathcal{N}(x|0, I)$ for $x \in \mathbb{R}$, to an approximate data distribution $p_1 \approx p_{\text{data}}(x)$. A vector field $u_t(x)$, which leads to an ordinary differential equation (ODE) $\frac{\mathrm{d}\phi_t(\mathbf{x})}{\mathrm{d}t} = u_t(\phi_t(\mathbf{x}))$, is used to generate the probability path via the push-forward equation,

$$p_t = [\phi_t]_* p_0 = p_0(\phi_t^{-1}(x)) \det \left[ \frac{\partial \phi_t^{-1}}{\partial x}(x) \right], \qquad (1)$$

which could be approximated with a trainable network $\hat{v}_t(x; \theta)$.

Due to the complexity of defining an appropriate $p_t$ and $u_t$, we could alternatively define a conditional probability path $p_t(x|x_1)$, which is usually derived through a conditional vector field $u_t(x|x_1)$ for each data point $x_1$ (Lipman et al., 2022). The conditional vector field is then approximated with a trainable network $\hat{v}_t(x; \theta)$. The conditional flow matching loss,

$$\mathcal{L}_{\text{CFM}}(\theta) = \mathbb{E}_{t, p_{\text{data}}(x_1), p_t(x|x_1)} \| \hat{v}_t(x; \theta) - u_t(x|x_1) \|, \qquad (2)$$

is proved to have identical gradients w.r.t. $\theta$ with $\mathcal{L}_{\text{FM}} = \mathbb{E}_{t,p_{\text{data}}(x)}||\hat{v}_t(x;\theta) - u_t(x)||$ (Lipman et al., 2022), which means the model can generate a marginal vector field by simply learning from the $x_1$-conditioned vector fields, without access to $p_t(x)$ and $u_t(x)$. After training, a neural ODE is obtained, ready for sampling from $p_0$ to $p_t$ with an ODE solver (Jardine, 2011).

## 4 METHOD

ATOMFLOW uses a unified *biotoken* representation to jointly generate protein and ligand structures by learning the distribution of token positions conditioned on a ligand chemical graph $\mathcal{G}$. Figure 2 illustrates the overall framework. We introduce a rectified flow on token positions $\mathbf{x} \in \mathbb{R}^{N \times 3}$ and approximate its vector field with an SE(3)-equivariant structure prediction network. In this section, we introduce the flow matching model in Section 4.1, the biotoken feature representation in Section 4.2, the structure prediction module in Section 4.3, and the training and inference procedures in Section 4.4. The overview of our method is illustrated in Figure 2.

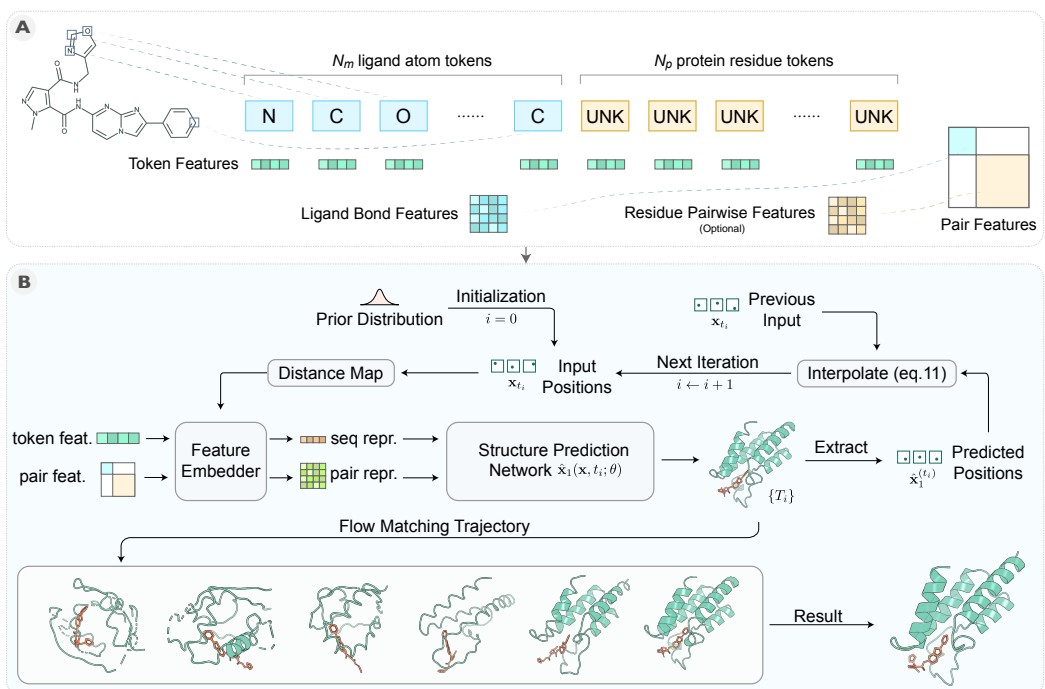

Figure 2: The inference process of ATOMFLOW. We represent the protein-ligand complex as a series of biotokens and embed their token and pair-level features (residue pairwise features are optional, see Appendix A.2). Starting from a noisy sample, the flow matching procedure gradually generates the designed structure $x_1$ with a structure prediction network.

### 4.1 FLOW MATCHING FOR PROTEIN-LIGAND COMPLEX GENERATION

For all types of tokens, we only consider their token positions to simplify the flow matching process. Thus, the positions of all tokens lie in the Euclidean space $\mathbb{R}^{N \times 3}$. Since a complex could be arbitrarily moved or rotated without changing its structure, we need to treat different coordinate representations as the same if they could be aligned with an SE(3) translation. Thus, every sample lies in the quotient space $Q : \mathbb{R}^{N \times 3}/\text{SE}(3)$. We define a rectified flow on Q using a conditional vector field

$$u_t(\mathbf{x} \mid \mathbf{x}_1) = \frac{1}{1-t}\Big(\text{align}_{\mathbf{x}}(\mathbf{x}_1) - \mathbf{x}\Big), \tag{3}$$

where $\mathbf{x}_1$ is the target structure from the data distribution, and $\text{align}_{\mathbf{x}}(\mathbf{x}_1)$ is its best root mean square deviation (RMSD) alignment to $\mathbf{x}$. We train the network to approximate $u_t(\mathbf{x} \mid \mathbf{x}_1)$ by minimizing $\mathcal{L}_{\text{CFM}}(\theta)$. With approximation (Appendix A.3), we find it more numerically stable to use an FAPE-based loss (Jumper et al., 2021), which does not change the final training target.

$$\mathcal{L}_{\text{CFM-FAPE}}(\theta) = \mathbb{E}_{t,p_{\text{data}}(\mathbf{x}_1),p_t(\mathbf{x}|\mathbf{x}_1)}\Big[\frac{1}{1-t}\text{FAPE}\big(\hat{\mathbf{x}}_1(\mathbf{x}, t; \theta), \mathbf{x}_1\big)\Big]. \tag{4}$$

Here, $\hat{\mathbf{x}}_1$ is predicted by our structure network. We partition the FAPE loss into protein-protein, protein-ligand, and ligand-ligand interactions with appropriate scaling factors.

## 4.2 REPRESENTATION OF CONDITIONAL FEATURES

The generation process of ATOMFLOW is conditioned on the ligand chemical graph $\mathcal{G}$ and a designated protein length $N_p$. We model such conditions as an additional condition to the vector field $u$. As a result, the inputs of the prediction network $\hat{\mathbf{x}}_1$ are augmented to accept conditional features. With the biotoken representation, we embed all such features as $f^{\text{token}}$ and $f^{\text{pair}}$ as illustrated in Figure 2A.

For a ligand chemical graph $\mathcal{G}$, we embed the chemical properties as $f^{\text{token}}$ of ligand tokens. The chemical bonds $\mathcal{E}$ are embedded in $f^{\text{pair}}$ as a multi-dimensional adjacency tensor, each dimension representing a bond type. For residue tokens, we embed the relative residue position (Shaw et al., 2018) in $f^{\text{pair}}$, while $f^{\text{token}}$ may represent other known conditions. We concatenate the protein and ligand features to form a unified feature tensor, eliminating the need to distinguish different types of tokens when processing the features. Further details are provided in Appendix A.2.

## 4.3 STRUCTURE PREDICTION NETWORK

The structure prediction network $\hat{\mathbf{x}}_1(\mathbf{x}, t; \theta)$ [2] predicts the token frames $\{T_i\}$, which can be used to extract token positions $\mathbf{x}_1$, given a series of noisy positions $\mathbf{x}$ at timestamp $t$. It encodes $\mathbf{x}$, along with $f^{\text{token}}$ and $f^{\text{pair}}$, with an SE(3) invariant encoding module, processing the representation with a transformer stack, and generates the predicted structure with a structure module based on invariant-point attention (IPA) (Jumper et al., 2021), as illustrated in Figure 2B. The network jointly processes two kinds of biotokens, protein residues and ligand atoms, with different spatial scales, and handles such differences with special care.

**Distance Map.** The input coordinates $\mathbf{x}$ are encoded by projecting the one-hot binned distance map between input coordinates for each token pair to the feature space

$$t_{i,j} = \text{Linear}(\text{BinRepr}(\|\mathbf{x}^{(i)} - \mathbf{x}^{(j)}\|)), \tag{5}$$

where the bins are not divided equally considering the different precision requirements between residues and atoms. This representation is SE(3) invariant, since the internal distance does not change under rigid transformation.[3]

**Feature Embedder.** The feature embedder generates a single representation $s \in \mathbb{R}^{N \times c_s}$ and pair representation $z \in \mathbb{R}^{N \times N \times c_z}$ from distance map $h$, noise level $t$, $f^{\text{token}}$ and $f^{\text{pair}}$ for the following steps. The noise level is encoded with Gaussian Fourier embedding (Song et al., 2021). The local features are concatenated and projected to single representation $s$ and pair representation $z$, $s_i = \text{Linear}(f_i^{\text{local}})$. The pair features and input encoding are projected and added to $z$

$$z_{i,j} = \text{Linear}(f_i^{\text{local}}) + \text{Linear}(f_j^{\text{local}}) + f_{i,j}^{\text{pair}} + t_{i,j}. \tag{6}$$

As described in Section 4.2, different token types can be treated the same and processed uniformly.

**Structure Module.** The structure module generates a predicted complex structure, represented as a series of token frames $T^N$. For ligand atoms, the rotation of the predicted frame is always identity rotation, while the translation equals its position. It first processes $z$ through a deep transformer stack (Appendix A.4) to obtain a denoised pair representation $z'$, and converts $s$ and $z'$ to $T^N$ through a series of shared-weight IPA block

$$T_{1\cdots N} = \text{IPAStack}(s_{1\cdots N}, \text{TransformerStack}(z_{1\cdots N, 1\cdots N})). \tag{7}$$

The IPA stack outputs a sequence of frames for each token, while the rotations for atom tokens are dropped and replaced with the atom frame demonstrated in Section 4.2. The final output represents the full complex structure $\hat{\mathbf{x}}$, while token positions $\hat{x}_1$ are calculated as previously described.

**Auxiliary Head.** We add an auxiliary head to predict the pairwise binned distance from the denoised pair representation $z'$, $h_i = \text{softmax}(\text{Linear}(z_i'))$, which directly supervises the input of structure module and has been proved to be helpful during training (Jumper et al., 2021). The bins are also unevenly divided to accommodate the multi-scale characteristics of the predicted complex.

---

[2] Though $\hat{\mathbf{x}}_1$ is a function of $\mathbf{x}, t, f^{\text{token}}, f^{\text{feat}}$, we omitted certain parameters to simplify the text.

[3] To accommodate the precision differences between ligands and proteins, the bin intervals are dense between 1Å (approximate length of a chemical bond) and 3.25Å (approximate distance between adjacent amino acids) and sparser beyond 3.25Å.

### 4.4 TRAINING AND INFERENCE

We train the network $\hat{\mathbf{x}}_1$ by sampling data points and timestamps, calculating the noisy input, and supervising the predicted results. At inference time, we transform the token positions sampled from the prior distribution through the predicted vector field with an ODE solver, and output the structure obtained at the final step.

**Training.** We sample the timestamp $t$ from the logit normal distribution, assigning more weight on intermediate steps, which helps the model to achieve better performance on hard timestamps (Esser et al., 2024; Karras et al., 2022). The prior distribution $q(x)$ is selected as $\mathcal{N}(0, \sigma_{\text{data}})$, where $\sigma_{\text{data}} = 10$. The input $\mathbf{x}$ is given by interpolating the data point and a sample from the prior distribution. The training procedure is shown in Algorithm 1.

**Inference.** A scheduler of noise levels $\{t_i\}_{i=0}^m, t_0 = 0, t_m = 1$ is used to determine the noise level $t_i$ of each sampling step $x_{t_i}$. Starting from a noisy sample $x_{t_i} = x_0$ as the initial model input, the structure prediction network predicts the vector field, which gives $x_{t_{i+1}}$ with the Euler's Method, i.e.

$$\mathbf{x}_{t_{i+1}} = \mathbf{x}_{t_i} + \frac{t_{i+1} - t_i}{1 - t_i} \left( \text{align}_{\mathbf{x}_{t_i}} \left( \text{Extract}(\hat{\mathbf{x}}_1(\mathbf{x}_{t_i}, t_i; \theta)) \right) - \mathbf{x}_{t_i} \right), \tag{8}$$

where the Extract function extracts the token positions from the predicted token frames. The model output at the last step is adopted as the final result. The inference procedure is shown in Algorithm 2.

---

**Algorithm 1** Training

**Require:** data distribution $p(\mathbf{x})$, prior distribution $q(\mathbf{x})$, trainable model parameters $\theta$
1: **while** not converged **do**
2:     sample complex structure $\mathbf{x}_1$ and its corresponding ligand chemical graph $\mathcal{G}$ from $p(\mathbf{x}), t \sim [0, 1), \mathbf{x}_0 \sim q(\mathbf{x})$
3:     $N, f^{\text{token}}, f^{\text{pair}} \leftarrow \text{Embedder}(\mathcal{G}, N_p)$
4:     $\mathbf{x}_t \leftarrow t \cdot \mathbf{x}_1 + (1 - t) \cdot \text{align}_{\mathbf{x}}(\mathbf{x}_0)$
5:     $\theta \leftarrow \text{Optimizer}(\theta, (\mathbf{x}_t, f^{\text{token}}, f^{\text{pair}}, t), \mathcal{L})$
6: **end while**
7: **return** $\theta$

**Algorithm 2** Inference

**Require:** Chemical graph $\mathcal{G}$, residue count $N_p$, scheduler $t_{0\cdots m}$, prior distribution $q(\mathbf{x})$, model parameters $\theta$
1: $N, f^{\text{token}}, f^{\text{pair}} \leftarrow \text{Embedder}(\mathcal{G}, N_p)$
2: sample token positions $\mathbf{x}_{t_0} \sim q(\mathbf{x})$
3: **for** $i = 0$ to $m - 1$ **do**
4:     $T_{1\cdots N} \leftarrow \hat{\mathbf{x}}_1(\mathbf{x}_{t_i}, f^{\text{token}}, f^{\text{pair}}, t_i; \theta)$
5:     $\hat{\mathbf{x}}_1 \leftarrow \text{Extract}(T)$
6:     calculate $\mathbf{x}_{t_{i+1}}$ as Equation 8
7: **end for**
8: **return** $T$

---

## 5 EXPERIMENTS

Following previous protein design models (Yim et al., 2023a; Lin & AlQuraishi, 2023; Watson et al., 2023) and binder design models (Krishna et al., 2024), we evaluate ATOMFLOW through in silico experiments on key metrics of our generated binder including self-consistency, binding affinity, diversity and novelty.

### 5.1 EXPERIMENT SETUP

**Training Data.** We train the denoising model on two datasets: PDBBind (Liu et al., 2017), a protein-ligand conformer dataset derived from the Protein Data Bank (PDB) (Berman et al., 2000), and SCOPe (Chandonia et al., 2022), a structure categorical dataset for protein. The model is first trained on solely generating the protein structure for 400k steps, and then finetuned on co-generating both the protein and ligand structure for 300k steps.

**Baseline and Model Variant.** We compare ATOMFLOW with the state-of-the-art binder generation method RFDiffusionAA (Krishna et al., 2024), which is extensively trained on almost all known data. Since RFDiffusionAA requires a fixed ligand structure at the binding state as input, we extend our method to work under its setting. For ATOMFLOW, besides the original setting (ATOMFLOW-N), we also train a version of our model with the pairwise distance matrix of the bound structure as an auxiliary hint input (ATOMFLOW-H). We also compare with a repurposed version of an AlphaFold 3 replication model Chai-1 (team et al., 2024) (see Appendix A.5 for details).[4] We exclude PocketGen (Zhang et al., 2024) and FlowSite (Stark et al., 2024) since they can only refine the pocket residues of a given binder. We discuss them with an additional experiment in the appendix.

---

[4]Due to license restrictions, we cannot use the original AlphaFold 3. In our preliminary experiments comparing Chai-1, Boltz (Wohlwend et al., 2024), and Protenix (Team et al., 2025), Chai-1 performed best and is used as a substitute in this work.

**Evaluation Set.** Following RFDiffusionAA, we evaluate all methods on the ligand set (evaluation set) from RFDiffusionAA (CCD code[5]: FAD, SAM, IAI, OQO). IAI and OQO are two ligands newly added to PDB, out of the training set of all methods. We conduct an *extended evaluation* on a larger set of 20 ligands (Appendix A.6). The extended evaluation set comprises ligands from inside and outside the training set, with both long and short lengths, and the trends remain consistent with those in the main text. We also include a speed test demonstrating ATOMFLOW is 5 times faster than RFDiffusionAA in Appendix A.6.

## 5.2 SELF-CONSISTENCY AND CONFORMER LEGITIMACY

In this section, we evaluate the generated protein structure by self-consistency RMSD and the predicted ligand structure at the binding state by detecting structural violence in the conformer. We further evaluate the legitimacy of designed complex conformer on geometric distribution and chemical validity.

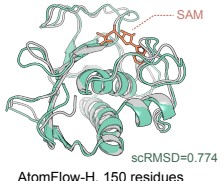 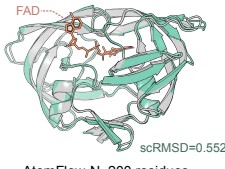 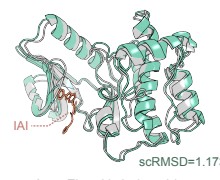 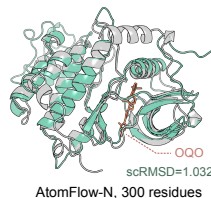

Figure 3: Designed structures for different ligands at different lengths. We align the ESMFold predicted structure to the designed structure, and report the scRMSD metric. Green: designed protein; Orange: designed ligand conformer; Grey: ESMFold predicted protein.

**Protein Structure.** For protein structures, self-consistency RMSD is widely adopted as a metric to evaluate their legitimacy (Lin & AlQuraishi, 2023; Watson et al., 2023), which compares the generated structure and the folding of its sequence predicted by an accurate model. We adopt LigandMPNN (Dauparas et al., 2023) to predict possible sequences from the generated structures. We first generate 8 sequences for all designed structures with LigandMPNN, then predict the corresponding protein structure with ESMFold (Lin et al., 2023), and the metric for each generated structure is calculated as the minimum rooted mean squared distance between the designed structure and predicted structure (scRMSD). For each ligand in the evaluation set, we generate 10 structures for lengths in [100, 150, 200, 250, 300]. The results are shown in Table 2. We illustrate several generated samples in Figure 3, and the cumulative distribution of scRMSD among them in Figure 4 and Figure S2B.

| Method | Overall | SAM | FAD | IAI | OQO |
|---|---|---|---|---|---|
| ATOMFLOW-H | **0.57** | **0.60** | 0.36 | **0.58** | **0.74** |
| ATOMFLOW-N | 0.50 | 0.50 | 0.38 | **0.58** | **0.54** |
| RFDiffusionAA | 0.52 | **0.60** | **0.58** | 0.48 | 0.42 |
| Chai-1 | 0.46 | 0.48 | 0.52 | 0.52 | 0.32 |
| RFDiffusion | 0.33 | 0.04 | 0.50 | 0.44 | 0.32 |

Table 2: Proportion of samples with scRMSD < 2 on the evaluation set (higher is better).

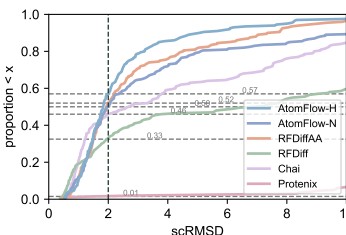

Figure 4: scRMSD distribution of samples on the evaluation set.

ATOMFLOW-H achieves the best overall performance on the evaluation set, ranking first on 3 out of 4 ligands. ATOMFLOW-N, the version without structural hints, also performs comparably to RFDiffusionAA, and even outperforms it on the out-of-distribution test cases IAI and OQO. Although not specifically designed for de novo generation, Chai-1 shows acceptable performance on protein structure tasks. The relatively limited performance of RFDiffusion is expected, as its strong binding potential for guiding protein-ligand interactions can lead to structural disruption. Including structural hints from the ligand conformer slightly improves binder quality, likely because the pocket shape is partially revealed through the input.

---

[5]The identifing three-letter code in the chemical components directory (Westbrook et al., 2015).

**Conformer Legitimacy.** We evaluate the chemical validity of the protein-ligand complex conformers with PoseBusters (Buttenschoen et al., 2024) and PoseCheck (Harris et al., 2023). We also evaluate the geometric distribution of common chemical bond lengths generated by ATOMFLOW in comparison to the ground truth bond lengths in our training set, as well as comparing the Ramachandran plots between ATOMFLOW-generated proteins and natural ones. We illustrate the results in Appendix A.6. The results indicate that ATOMFLOW generates legitimate samples with metrics close to those of the natural proteins.

### 5.3 BINDING AFFINITY

In this section, we evaluate binding affinity using two in silico metrics, acknowledging that such computational estimates can only serve as proxies. Ultimately, wet-lab experiments remain the gold standard for validating binding affinity, though they are typically too costly for large-scale benchmarking.

We first report the **AutoDock Vina score** (Eberhardt et al., 2021), a widely adopted docking-based energy metric (Zhang et al., 2024). For each designed binder, we pack side chains using the Rosetta packer (Leaver-Fay et al., 2011), and report the minimum docking score across all generated conformations. This score serves as an approximate measure of interaction energy between the ligand and the designed protein.

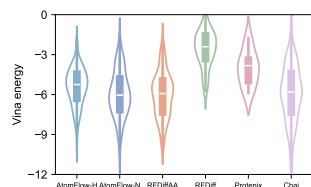

Figure 5: Vina score distribution on the evaluation set.

As a complementary signal, we compute the **min PAE interaction (`min_ipAE`)**—a metric derived from the Predicted Aligned Error (PAE) matrix of an AlphaFold 3-like model. Specifically, `min_ipAE` reflects the model's confidence in the relative positioning between ligand and protein residues; lower values indicate higher structural confidence in the binding interface. Although originally introduced for protein–protein interactions (Zambaldi et al., 2024), this metric has demonstrated strong correlation with binder quality. To obtain `min_ipAE`, we input LigandMPNN-designed sequences and their respective ligands into Chai-1 (team et al., 2024), and extract the lowest interaction PAE value between protein and ligand tokens, following the AlphaProteo protocol.

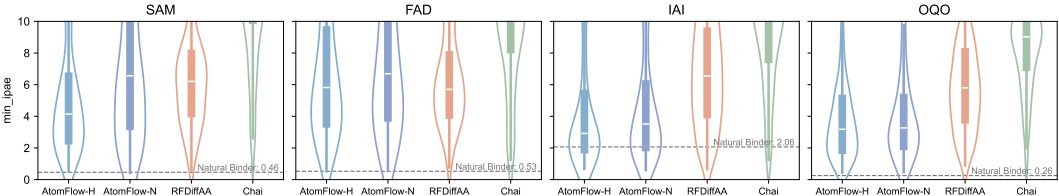

Figure 6: min PAE interaction (`min_ipAE`) of samples on the evaluation set (lower is better) predicted by Chai-1. The value of natural binder in PDB is highlighted.

As shown in Figure 6, binders generated by ATOMFLOW-H achieve lower `min_ipAE` than those from RFDiffusionAA on 3 of 4 ligands. ATOMFLOW-N performs slightly worse than ATOMFLOW-H but still shows strong results, without structural hint. Both ATOMFLOW and RFDiffusionAA produce results comparable to natural complexes. The AutoDock Vina results, shown in Figure 5 and detailed in Figure S2A, indicates that ATOMFLOW also matches or exceeds the performance of RFDiffusionAA in terms of docking energy, aligning with the trends observed in the `min_ipAE` metric. Similar results are observed on the extended evaluation set (Figure S2C,D).

We further compare ATOMFLOW with RFDiffusionAA in a realistic setting where the bound conformer is unknown. We set the target ligand as luminespib (PDB id: 2GJ), an Hsp90 inhibitor (Piotrowska et al., 2018). A designed protein binder for luminespib may act as a protein drug carrier to enhance drug efficacy. Luminespib is a molecule ligand with 33 heavy atoms, so that the conformer is quite flexible when docked to different receptors. We design 10 binders for luminespib using ATOMFLOW and RFDiffusionAA. The ideal conformer from PDB is provided to RFDiffusionAA, while no conformer is provided to ATOMFLOW. The binding energy of the designed structures and one designed sample with PLIP (Adasme et al., 2021) to demonstrate the protein-ligand interaction are illustrated in Figure 7. It is shown that ATOMFLOW generates more binders with higher binding affinity than RFDiffusionAA, and significantly outperforms RFDiffusionAA on the lowest energy

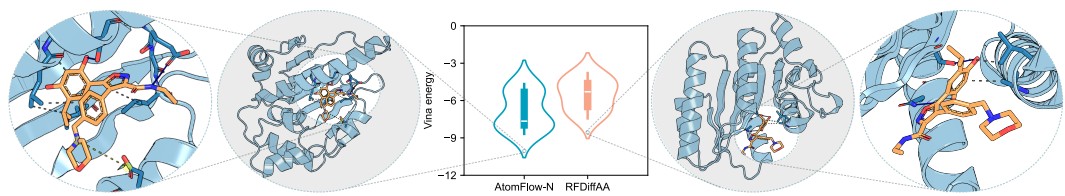

Figure 7: ATOMFLOW-N designs binders with lower vina energy distribution than RFDiffusionAA on 2GJ without the bound structure. Illustrations of one sample for each method with PLIP demonstrates that the ATOMFLOW-N designed binder has more chemical interactions with the ligand.

among all generated structures. This demonstrates that a proper bound structure is crucial to the performance of RFDiffusionAA, while ATOMFLOW does not rely on such structure and generates proper conformers by co-modeling the structure space of proteins and ligands.

### 5.4 DIVERSITY AND NOVELTY

In this section, we report the diversity and novelty of ATOMFLOW, following common practice in literature (Krishna et al., 2024; Yim et al., 2023b). Diversity refers to the structural divergence of the designed binders for a certain ligand, while novelty refers to how close a designed protein is to the known proteins. For diversity, we generate 100 structures with 200 residues for each ligand, and then use MaxCluster (Herbert, 2008) to calculate the pairwise structural distance (TMScore) of the outputs and report the number of clusters using different thresholds of the maximum distance within a cluster. For novelty, we generate 4 structures with residue count in $[100, 101, \cdots, 300]$ for each ligand, and then calculate the highest TM-score (Zhang, 2005) between a designed structure and any similar structure searched by FoldSeek (Kempen et al., 2024) (pdbTM), as well as the protein scRMSD. The search range of pdbTM is all known protein structures in PDB. Results of RFDiffusionAA are provided in Figure S6 (Appendix A.6).

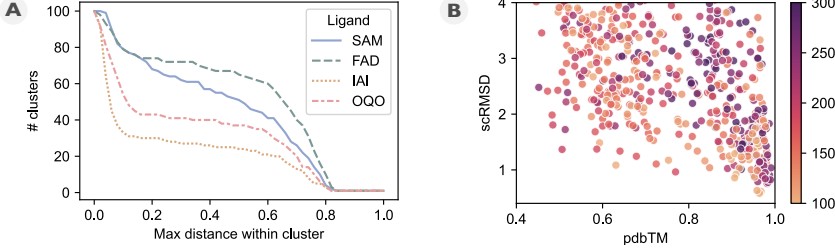

Figure 8: **A**: Cluster count based on different thresholds of the maximum difference (TMScore) within the cluster for each ligand in the evaluation set. ATOMFLOW generates diverse binder folds for all ligands, not restricted to the existing binder structure. **B**: Scatter plot of designability (scRMSD) vs. novelty (pdbTM) for ligands in the evaluation set. ATOMFLOW successfully designs self-consistent structures with high pdbTM, demonstrating high novelty.

Figure 8A shows that ATOMFLOW produces diverse structures across ligands, with variability depending on the ligand. Incorporating protein-only data during training helps the model capture structural patterns beyond known complexes. As shown in Figure 8B, most generated designable folds remain close to known ones, and the degree of novelty is lower than RFDiffusionAA, likely due to smaller training scale, consistent with previous reports (Huguet et al.).

## 6 CONCLUSION AND FUTURE WORK

We propose ATOMFLOW, a de novo protein binder design method for small molecule ligands that explicitly models ligand flexibility without requiring a fixed conformer. By representing protein–ligand complexes as unified biotokens and applying an SE(3)-equivariant flow matching framework, ATOMFLOW achieves comparable or superior binder design quality to RFDiffusionAA while offering faster inference and robustness when the ligand binding conformation is unknown. We further introduce an experimentally validated binding affinity metric for comprehensive evaluation. Future directions

include enabling finer structural control, scaling up training, and extending ATOMFLOW to broader biomolecules such as DNA, RNA, and metal ions.

## REPRODUCIBILITY STATEMENT

We include the source code of the AtomFlow model and its corresponding checkpoint with a ready-to-use Gradio interface in the supplementary materials. Instructions for setting up the environment and launching the web-based interface are provided as a README file. Further details on the model implementation and training are available in Appendix A.4

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

# A APPENDIX

## A.1 PROTEIN FRAMES

Proteins are composed of amino acid chains linked by peptide bonds, forming a backbone with protruding side chains. Each amino acid's position and orientation are described by a local coordinate system, or protein frame, centered on three key backbone atoms: the alpha carbon (C$\alpha$), the carbonyl carbon (C), and the amide nitrogen (N). These atoms act as reference points for establishing the frame. The alpha carbon (C$\alpha$) typically acts as the origin. The vector from C$\alpha$ to the amide nitrogen (N) is normalized to define one axis of the frame. A second axis is defined by the normalized vector from C$\alpha$ to the carbonyl carbon (C). The third axis is formed by the cross product of these two vectors, creating an orthogonal, right-handed coordinate system. The residue frame is typically represented as an SE(3) transformation $T = (R, t)$, which maps a vector from this local system to the global coordinate system. In this transformation, $t$ corresponds to the position of C$\alpha$ in the global system, and $R$ represents the rotation needed to align the residue's structure within the global context.

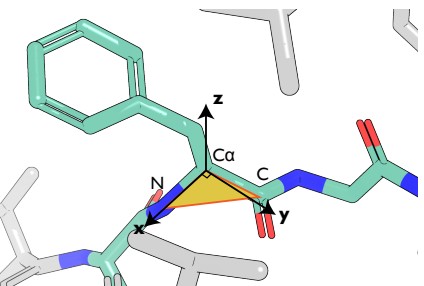

Figure S1: A protein frame illustration. The C$\alpha$, C, N atoms form a panel, which is the xy panel. The x-axis is defined as the orientation from C$\alpha$ to N, while the y-axis is on the panel and perpendicular to the x-axis. The z-axis is perpendicular to the xy panel.

## A.2 DETAILS ON BIOTOKENS

**Token Features.** For ligand atom tokens, the token-level feature set includes: chirality, degree, formal charge, implicit valence, number of H atoms, number of radical electrons, orbital hybridization, aromaticity, and ring size. The pair-level feature is provided as one-hot embedding of the bond type. For residue tokens, no token-level feature is known, while the pair-level features are optional and may contain the binned distance of residue index between residues for fully de novo generation (our experiment), and motif structures can be provided as additional pairwise features when needed. All features are encoded as a one-hot vector and concatenated.

**Token Frames.** The final loss we adopted $\mathcal{L}_{\text{CFM-FAPE}}$ requires aligning the predicted structure to the local frame of every token. The frames of protein residues can be naturally defined as in Section 3. However, the frames of ligand atoms could not be chosen directly. Since a frame could be calculated from the coordinate of 3 atoms, we need to choose an atom triplet for every atom token.

We first obtain a canonical rank of every atom that does not depend on the input order (Schneider et al., 2015). The atoms are then renamed to their rank. For atoms $x$ with a degree greater than or equal to 2, we select the lexicographically smallest triplet $(u, x, v)$ to define the frame, where $u$ and $v$ are neighbors of $x$. For atoms with a degree of 1, $u$ is the only neighbor of $x$, and $v$ is chosen as one of $u$ 's neighbors. This method ensures that each atom's frame is defined in a consistent manner, irrespective of its position in the input sequence, thereby facilitating the model to learn a consistent structural target.

**Extending Token Types and Features.** Though ATOMFLOW only considers the interaction between protein and molecule ligands, the unified biotoken has the potential to extend to all biological entities, including DNA, RNA, etc, by defining the token position, token frame, local and pair features, and the representation of the internal structure. For example, an RNA can be represented as a sequence of nucleotides, with the token position defined as its mass center, and the token frame calculated from an atom triplet, such as C2-N1-C6.

The token features can also be extended to support more types of known information. For example, the local features could also contain an embedding to indicate the preferred secondary structure, or whether a ligand atom is required to be closer to the designed protein; the pair features could also contain the motif information with a distance map.

### A.3 Details on the Flow Matching Process

For all types of tokens, we only consider their token positions to simplify the flow matching process. Thus, the positions of all tokens lie in the Euclidean space $\mathbb{R}^{N \times 3}$. Since a complex could be arbitrarily moved or rotated in the coordinate space without changing its structure, we need an algorithm that treats different position series as the same if they could be aligned with an SE(3) translation. Thus, every data point we consider now lies in the quotient space $\mathbb{R}^{N \times 3}/\text{SE}(3)$. This quotient space is proved to be a Riemannian manifold (Diepeveen et al., 2024).

For a Riemannian manifold, the flow matching process could be defined using a premetric (Chen & Lipman, 2024). A premetric $d : \mathcal{M} \times \mathcal{M} \to \mathbb{R}$ should satisfy: 1. $d(x, y) \geq 0$ for all $x, y \in \mathcal{M}$; 2. $d(x, y) = 0$ iff $x = y$; 3. $\nabla d(x, y) \neq 0$ iff $x \neq y$.

We define our premetric as the minimum point-wise rooted sum of squared distance (RMSD) among all pairs of possible structures in the original space $\mathbb{R}^{N \times 3}$ for two elements in the quotient space $d(x, y) = \|\text{align}_x(y) - x\|$, which satisfies all three conditions.

*Proof.* Since the premetric is defined as a norm, it satisfies condition 1 by nature. When $x = y$, the best alignment that aligns $y$ to $x$ could derive the exact same position as $x$, yielding a zero norm. When $x \neq y$, when $y$ is aligned to $x$, there's still a structural difference between the structures, thus the premetric is not zero. For condition 3, by defining $y' = \text{align}_x(y)$, we have

$$\nabla d(x, y) = \nabla \sqrt{\sum_{i=1}^{n} (y'_i - x_i)^2} = \frac{y' - x}{\|y' - x\|} = \frac{\text{align}_x(y) - x}{\|\text{align}_x(y) - x\|} \geq 0. \tag{9}$$

Thus $d(x, y)$ satisfies all the conditions as a qualified premetric. $\qquad \square$

With such premetric, and a monotonically decreasing differentiable scheduler $\kappa(t) = 1 - t$, we could obtain a well-defined conditional vector field that linearly interpolates between the noisy and real data (Chen & Lipman, 2024)

$$u_t(x|x_1) = \frac{d \log \kappa(t)}{dt} d(x, x_1) \frac{\nabla d(x, x_1)}{\|\nabla d(x, x_1)\|^2} = \frac{1}{1 - t}(\text{align}_x(x_1) - x). \tag{10}$$

The vector field in equation 10 is calculated by substituting equation 9 into the left side. This vector field provides the direction for moving straight towards $x_1$, and generates a probability flow that interpolates linearly between noisy sample $x_0$ and data sample $x_1$.

Since the vector field is defined as a function of $x_1$, we could learn the vector field with a structure prediction model $\hat{x}_1(x, t; \theta)$. By substituting equation 10 into equation 2, we obtain the training loss

$$\mathcal{L}_{\text{CFM}}(\theta) = \mathbb{E}_{t, p_{\text{data}}(x_1), p_t(x|x_1)} \left\| \frac{1}{1 - t}(\text{align}_x(\hat{x}_1(x, t; \theta)) - \text{align}_x(x_1)) \right\|. \tag{11}$$

**Loss Function** $\mathcal{L}_{\text{CFM}}$ calculates an aligned RMSD by aligning $\mathbf{x_1}$ and $\hat{\mathbf{x}_1}$ to $x$, while the FAPE loss calculates an averaged RMSD by aligning $\hat{\mathbf{x}_1}$ to each residue frame of $\mathbf{x_1}$, which could be extended to the token frame (Appendix A.2). Let $\text{align}_{x,i}(y)$ denote aligning $y$ to the $i$-th token frame of $x$, we

have

$$\mathcal{L}_{\text{CFM}} = \mathbb{E}_{t,p_{\text{data}}(x_1),p_t(x|x_1)} \left\| \frac{1}{1-t} (\text{align}_x(\hat{x_1}(x,t;\theta)) - \text{align}_x(x_1)) \right\|$$

$$\approx \mathbb{E}_{t,p_{\text{data}}(x_1),p_t(x|x_1)} \left\| \frac{1}{1-t} \cdot \frac{1}{N} \sum_{i=1}^{N} \left( \text{align}_{x,i}(\hat{x_1}(x,t;\theta)) - \text{align}_{x,i}(x_1) \right) \right\|$$

$$\approx \mathbb{E}_{t,p_{\text{data}}(x_1),p_t(x|x_1)} \left\| \frac{1}{1-t} \cdot \frac{1}{N} \sum_{i=1}^{N} \left( \text{align}_{x_1,i}(\hat{x_1}(x,t;\theta)) - \text{align}_{x_1,i}(x_1) \right) \right\|$$

$$\approx \mathbb{E}_{t,p_{\text{data}}(x_1),p_t(x|x_1)} \left\| \frac{1}{1-t} \cdot \frac{1}{N} \sum_{i=1}^{N} \left( \text{align}_{x_1,i}(\hat{x_1}(x,t;\theta)) - x_1 \right) \right\|$$

$$= \mathcal{L}_{\text{CFM-FAPE}}$$

**Proposition 1.** $align_{\mathbf{x_1}}(\hat{\mathbf{x_1}}) = \mathbf{x_1} \iff \mathcal{L}_{\text{CFM}} = 0 \iff \mathcal{L}_{\text{CFM-FAPE}} = 0.$

*Proof.* When $\text{align}_{\mathbf{x_1}}(\hat{\mathbf{x_1}}) = \mathbf{x_1}$, we have $\forall i$, $\text{align}_{\mathbf{x_1},i}(\hat{\mathbf{x_1}}) = \mathbf{x_1}$. As a result, $\mathcal{L}_{\text{CFM}} = \mathcal{L}_{\text{CFM-FAPE}} = 0$. This establishes that:

$$\text{align}_{\mathbf{x_1}}(\hat{\mathbf{x_1}}) = \mathbf{x_1} \iff \mathcal{L}_{\text{CFM}} = 0 \quad \text{and} \quad \text{align}_{\mathbf{x_1}}(\hat{\mathbf{x_1}}) = \mathbf{x_1} \iff \mathcal{L}_{\text{CFM-FAPE}} = 0. \quad (12)$$

Now, assume $\mathcal{L}_{\text{CFM}} = 0$. Suppose $\text{align}_{\mathbf{x_1}}(\hat{\mathbf{x_1}}) \neq \mathbf{x_1}$. Then for all transformations $R$ and $t$, we have $R\hat{\mathbf{x_1}} + t \neq \mathbf{x_1}$, which implies: $\|\text{align}_{\mathbf{x_1}}(\hat{\mathbf{x_1}}) - \mathbf{x_1}\| \neq 0$, leading to $\mathcal{L}_{\text{CFM}} \neq 0$. This is a contradiction. Therefore, $\text{align}_{\mathbf{x_1}}(\hat{\mathbf{x_1}}) = \mathbf{x_1}$. This proves that

$$\mathcal{L}_{\text{CFM}} = 0 \iff \text{align}_{\mathbf{x_1}}(\hat{\mathbf{x_1}}) = \mathbf{x_1}. \quad (13)$$

Similarly, assume $\mathcal{L}_{\text{CFM-FAPE}} = 0$. Suppose $\text{align}_{\mathbf{x_1}}(\hat{\mathbf{x_1}}) \neq \mathbf{x_1}$. Then: $\|\text{align}_{\mathbf{x_1},i}(\hat{\mathbf{x_1}}) - \mathbf{x_1}\| \neq 0$, which leads to $\mathcal{L}_{\text{CFM-FAPE}} \neq 0$, again a contradiction. Therefore, $\text{align}_{\mathbf{x_1}}(\hat{\mathbf{x_1}}) = \mathbf{x_1}$. This proves that:

$$\mathcal{L}_{\text{CFM-FAPE}} = 0 \iff \text{align}_{\mathbf{x_1}}(\hat{\mathbf{x_1}}) = \mathbf{x_1}. \quad (14)$$

The proposition is proved by combining equation 12,13,14. $\qquad\square$

This means that both $\mathcal{L}_{\text{CFM}}$ and $\mathcal{L}_{\text{CFM-FAPE}}$ provide an optimization direction towards minimizing the SE(3) invariant structural difference between the predicted structure and the ground truth structure. Thus, we adopt $\mathcal{L}_{\text{CFM-FAPE}}$ as a realistic approximation of $\mathcal{L}_{\text{CFM}}$ and adopt it as the training objective during evaluation.

We divide the FAPE loss into protein-protein interaction, protein-ligand interaction, ligand-ligand interaction, and assign different $Z$s for the three parts. For the auxiliary head, we adopt the cross-entropy loss averaged over all token pairs for the predicted distance. The final training loss

$$\mathcal{L} = \alpha_1 \mathcal{L}_{\text{CFM-FAPE-pp}} + \alpha_2 \mathcal{L}_{\text{CFM-FAPE-pl}} + \alpha_3 \mathcal{L}_{\text{CFM-FAPE-ll}} + \alpha_4 \mathcal{L}_{\text{aux}}. \quad (15)$$

## A.4 DETAILS ON THE PREDICTION NETWORK

**Structure Module Specifications.** The main components of the structure module are derived from Alphafold 2 (Jumper et al., 2021), while our implementation builds on top of the widely acknowledged reimplementation OpenFold (Ahdritz et al., 2024). The TransformerStack consists of 14 layers of simplified Evoformer block, and the IPAStack consists of 4 layers of Invariant Point Attention (IPA) blocks. The MSA operations in the Evoformer block are simplified by replacing the operations on the MSA feature matrix with the single representation $s_i$. The weights of the IPA blocks are shared, and the structural loss is calculated on the outputs of each block and averaged.

**Training Details.** During training, we equally sample data from the SCOPe dataset (v2.08) and the PDBBind dataset (2020). We simply drop the data with more than 512 tokens, and we don't crop the filtered complexes since the cutoff is large enough and only filters out a relatively small portion of the data. We train our model on 10 NVIDIA RTX 4090 acceleration cards, with a batch size set to 10, which means the batch size on each device is set to 1. We use the Adam Optimizer (Kingma,

2014) with a weight-decaying learning rate scheduler, starting from $10^{-3}$ and decays the learning rate by 0.95 every 50k steps. We separate the training process into two stages: 1) initial training, $\alpha_1 = 0.5, \alpha_4 = 0.3, \alpha_2 = \alpha_3 = 0$; 2) finetuning, $\alpha_1 = \alpha_2 = \alpha_3 = 0.5, \alpha_4 = 0.3$.

Ligand tokens are not given during the first training stage. The first stage trains an unconditional protein generation model, while the second stage turns it to a conditional protein binder and ligand conformer generation model. The FAPE loss is defined as an average of all pairs of tokens in the original paper, so the calculation process first yield a FAPE matrix and then produce the average value of the matrix. The protein-protein, protein-ligand and ligand-ligand loss calculates the average value of the sub-matrixs defined as (row: protein, col: protein), (row: protein, col: ligand), and (row: ligand, col:ligand).

Since training a protein design model is significantly time-consuming, the design choices of our training strategy are largely determined by grid searching possible design space and we save the training trajectory of the first 30∼50k steps. We compare the training trajectories and select the best configuration that meets the following criteria: a) The final distogram loss should be close to the minimum we get among the configurations (around 2.0). b) The $\mathcal{L}_{\text{CFM-FAPE}}$ should not decline too fast at the first 10k steps. The first 10k steps are for the transformer stack to learn a relatively steady output, indicated by the decline of the distogram loss. A decline of $\mathcal{L}_{\text{CFM-FAPE}}$ at this stage will result in an undesired local minimum. Then $\mathcal{L}_{\text{CFM-FAPE}}$ should decline fast right after the distogram loss turns to decline much smoother. We select the configuration with the lowest $\mathcal{L}_{\text{CFM-FAPE}}$ at the end of training.

We decide the end of each training stage when the training converges, with the following criteria: a) the decline rate of every single loss is small. b) the structural violence of sampled structures (counts of CA atom violation) converges.

An initial study on directly training the second stage shows unsatisfactory training trajectory. Since the ligand conformer is way easier to generate compared to protein folds, the FAPE loss declines too fast even before the distogram loss, resulted in unstable TransformerStack output, and leading to a diverge of the model after around 30k steps. The resulted model with minimum loss is able to predict the ligand structure, with random protein residue position, which is unusable.

### A.5 EVALUATION DETAILS

**Specifications.** Following RFDiffusionAA, we use FAD, SAM, IAI, and OQO as the selected evaluation set. FAD and SAM are witnessed by both models as training data, while IAI and OQO are not, demonstrating the generalization ability. To further investigate the performance of our method, we conduct experiments on an extended set of 20 ligands (ligands from PDB id 6cjs, 6e4c, 6gj6, 5zk7, 6qto, 6i78, 6ggd, 6cjj, 6i67, 6iby, 6nw3, 6o5g, 6hlb, 6efk, 6gga, 6mhd, 6i8m, 6s56, 6tel, and 6ffe). The extended dataset includes ligand sizes (including hydrogen) ranging from 21 to 104 in length.

**Extended Set.** We illustrate the designability (scRMSD) and binding affinity (Vina energy) of ATOMFLOW-N in Figure S2. The extended evaluation shows that the performance of ATOMFLOW on the extended set is similar to the evaluation set shown in the main article, and demonstrates that ATOMFLOW is able to tackle almost all kinds of ligands.

**Repurposing Chai-1 for Structure Generation.** To enable protein design with Chai-1, we replace the protein sequence with $N$ UNK tokens as a placeholder chain and use the ligand SMILES string as the second chain. This allows Chai-1 to treat the input as a protein–ligand complex and generate 3D structures accordingly. No model weights or architecture were changed; only the input formatting was adapted. Despite not being trained for design, Chai-1 can produce reasonable structures under this setup; however, the resulting protein–ligand interfaces often lack clear or meaningful binding patterns.

### A.6 ADDITIONAL RESULTS

**Speed Comparasion with RFDiffusionAA.** We conducted experiments to generate samples for the ligand FAD using AtomFlow and RFAA, with amino acid lengths of 100, 150, 200, 250, and 300. For each length, we measured the time (in minutes) required to generate a single sample. Each

experiment was repeated three times, and we reported the average time along with the standard error. During the experiments, each method had exclusive access to its respective GPU.

| | $L = 100$ | $L = 150$ | $L = 200$ | $L = 250$ | $L = 300$ |
|---|---|---|---|---|---|
| AtomFlow | $0.49 \pm 0.0$ | $0.51 \pm 0.01$ | $0.58 \pm 0.01$ | $0.79 \pm 0.01$ | $0.88 \pm 0.0$ |
| RFAA | $2.48 \pm 0.03$ | $2.75 \pm 0.04$ | $3.23 \pm 0.01$ | $4.02 \pm 0.04$ | $4.58 \pm 0.05$ |
| Speedup | 5.06x | 5.39x | 5.57x | 5.09x | 5.2x |

Table 3: Comparison of AtomFlow and RFAA at different sequence lengths $L$

**Extended Evaluation.** We compare AtomFlow-N and RFDiffusionAA on the extended set of 20 ligands. The results are shown in Figure S2. ATOMFLOW matching or surpassing the performance of RFDiffusionAA on self-consistency RMAD, Vina energy and `min_ipAE`.

**Discussion on Pocket Design Models.** While the pocket design models address ligand-protein interactions, their focus is limited to refining pocket residues within a predefined radius. They lack the capacity to design full protein folds, making direct comparison with AtomFlow infeasible. We conducted an unfair experiment with PocketGen by providing a template binder to it, as detailed in Tabel 4. Despite this, the results demonstrate that AtomFlow consistently outperforms PocketGen in terms of fold quality across all radii.

For this experiment, we used the natural binders of four ligands—FAD (7bkc), SAM (7c7m), IAI (5sdv), and OQO (7v11)—as input. To evaluate the design capability of PocketGen (PG) under different constraints, we progressively increased the design radius (minimum distance to ligand) from 3.5 to 9.5. The masked target area expanded with the radius, requiring the model to redesign increasingly larger regions of the protein. When the radius exceeded the protein's dimensions, all residues were masked, simulating our full-design setting. The table below presents the min/median scRMSD values for designs generated by PocketGen at each radius. For reference, scRMSD < 2 is generally considered a successful design. Notably, PocketGen's performance deteriorated significantly as the radius increased, reflecting its reliance on template residues. At radius=8 for OQO, PocketGen generated designs with several residues misaligned with the ligand, leading to abnormally high scRMSD values. PocketGen does not support radius settings beyond 10, preventing direct simulation of the fully template-free design scenario. The results of ATOMFLOW are from our main experiment.

| Ligand | AtomFlow ($r=\infty$) | PG ($r$=3.5) | PG ($r$=5) | PG ($r$=6.5) | PG ($r$=8) | PG ($r$=9.5) |
|---|---|---|---|---|---|---|
| FAD | 0.79/3.74 | 7.10/7.38 | 6.75/7.81 | 7.29/8.35 | 20.92/24.23 | 23.12/25.23 |
| SAM | 0.83/2.01 | 2.12/2.62 | 2.77/2.99 | 2.94/4.03 | 12.39/14.49 | 13.79/14.74 |
| IAI | 0.56/1.82 | 0.71/0.85 | 0.95/1.02 | 2.04/2.28 | 3.59/5.53 | 9.02/11.71 |
| OQO | 0.59/1.63 | 1.20/1.26 | 1.70/1.79 | 2.40/2.45 | 11.59/11.94 | 2.13/2.41 |

Table 4: Comparison of AtomFlow and PocketGen (PG) on ligand-protein design tasks. For each ligand, we report the minimum/median scRMSD (in Å) of designed structures. AtomFlow results are shown for the full-design setting (r = inf), while PocketGen results are shown for increasing design radii (r = 3.5 to 9.5). Lower scRMSD indicates better structural accuracy. PocketGen performance degrades as the design radius increases, highlighting its reliance on template residues.

**Geometrical Distributions of Generated Structures.** We evaluated the common chemical bond length generated by AtomFlow vs. the ground truth bond length in our training set. Results shown in Figure S3 demonstrate that the AtomFlow generated ligands have similar geometric distribution to ground truth. We further evaluated the generated structures by plotting the Ramachandran plots. Results shown in Figure S4 suggests that the proteins generated by AtomFlow effectively capture the key structural characteristics of natural proteins.

**Chemical Validity of Generated Structures.** We use PoseBusters to evaluate the ligand conformer quality of AtomFlow-generated structures. On the extended evaluation set, 71.8% of conformers generated by AtomFlow-H pass all 19 tests (98.8% pass at least 18), whereas 51.4% of conformers from AtomFlow-N pass all 19 tests (82.3% pass at least 18). Since RFDiffusion-AA takes a fixed ligand conformer as input and outputs it without modification, so its score is entirely determined by the input and does not reflect model behavior.

We also use PoseCheck, a toolkit developed as part of a benchmark for structure-based drug design. Results are shown in Figure S5. While our primary objective is to develop a ligand-binding protein, instead of drug design, we find their metrics valuable for assessing the interaction and chemical validity of the protein-ligand complex. We report the following three metrics:

*Clash (lower is better)* evaluates the plausibility of protein-ligand binding poses by measuring the number of atomic pairs within a distance smaller than their van der Waals radii.

*Strain (lower is better)* assesses ligand conformational plausibility by calculating the difference in internal energy before and after ligand relaxation.

*Interactions (higher is better)* quantifies the number of chemical interactions formed in protein-ligand complexes, focusing on four types: Hydrogen Bond Acceptors, Hydrogen Bond Donors, Van der Waals Contacts, and Hydrophobic Interactions. Hyrophobic interactions and Van der Waals Contacts are illustrated separately.

We also report the PoseBusters results for AtomFlow-generated ligands on the extended evaluation set: For AtomFlow-H, 71.8% pass all 19 tests; 98.8% pass $\geq 18$. For AtomFlow-N: 51.4% pass all 19 tests; 82.3% pass $\geq 18$.

**Diversity and Novelty Results of the Baseline.** We conducted the diversity and novelty experiment on RFDiffusion-AA with the same configuration as our results reported in the main text. The results are shown in Figure S6. The diversity of AtomFlow designs is better than RFDiffusion-AA, while the AtomFlow generated results tend to be more conservative in terms of pdbTM novelty. We believe this is because we didn't train AtomFlow on a full training set including all PDB structures and the distillation data. This is our future work and we'll release an updated model once available.

## A.7   LLM USAGE

We use large language models (LLMs) for polishing writing, performing grammar checks, and implementing various utility scripts. LLMs made no contribution to experimental data analysis or the generation of research ideas. All outputs produced by LLMs were carefully reviewed and verified by the authors.

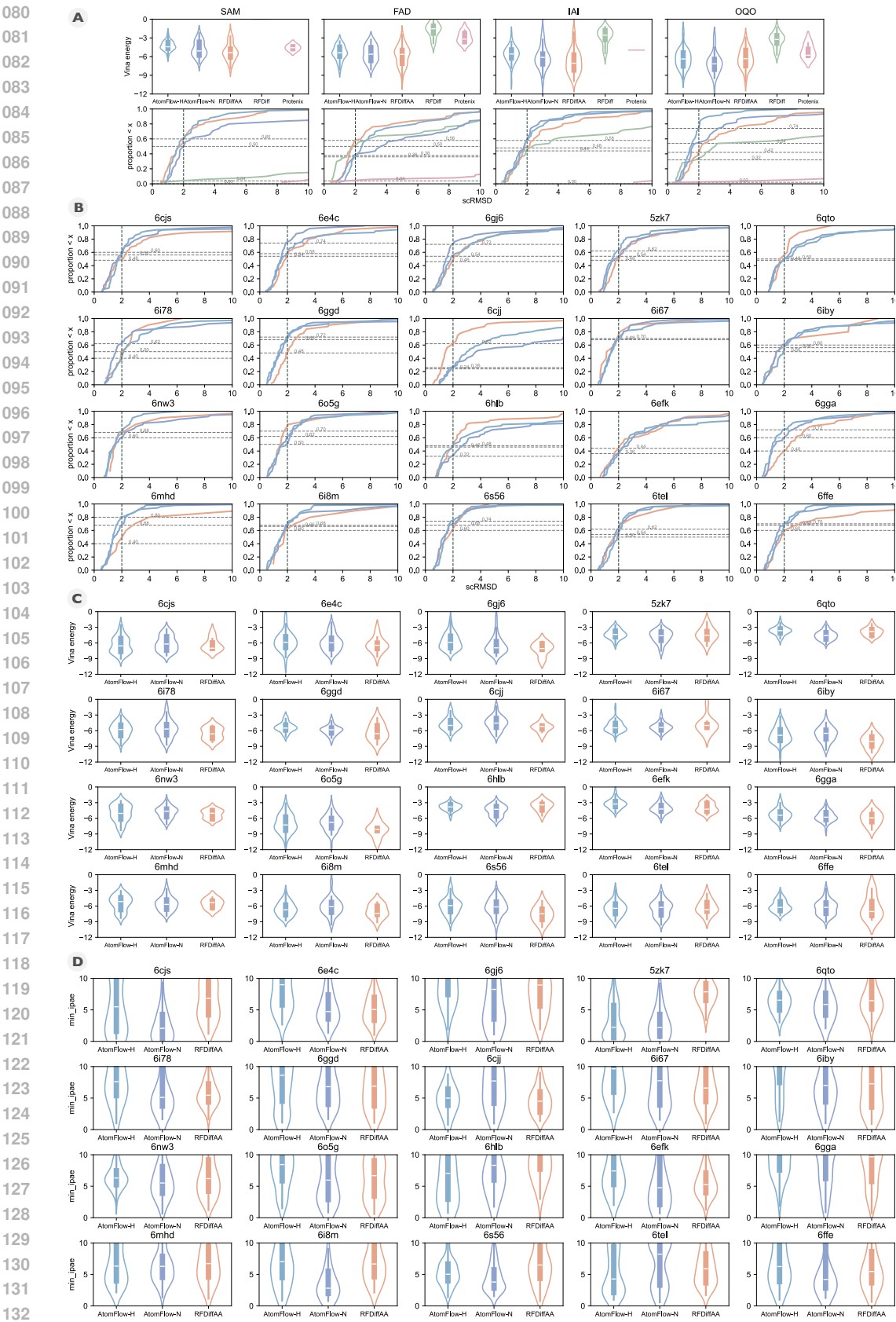

Figure S2: A: scRMSD and Vina energy of designs for the evaluation set; B: scRMSD of designs for the extended set; C: Vina energy of designs for the extended set; D: `min_ipAE` results for the extended set.

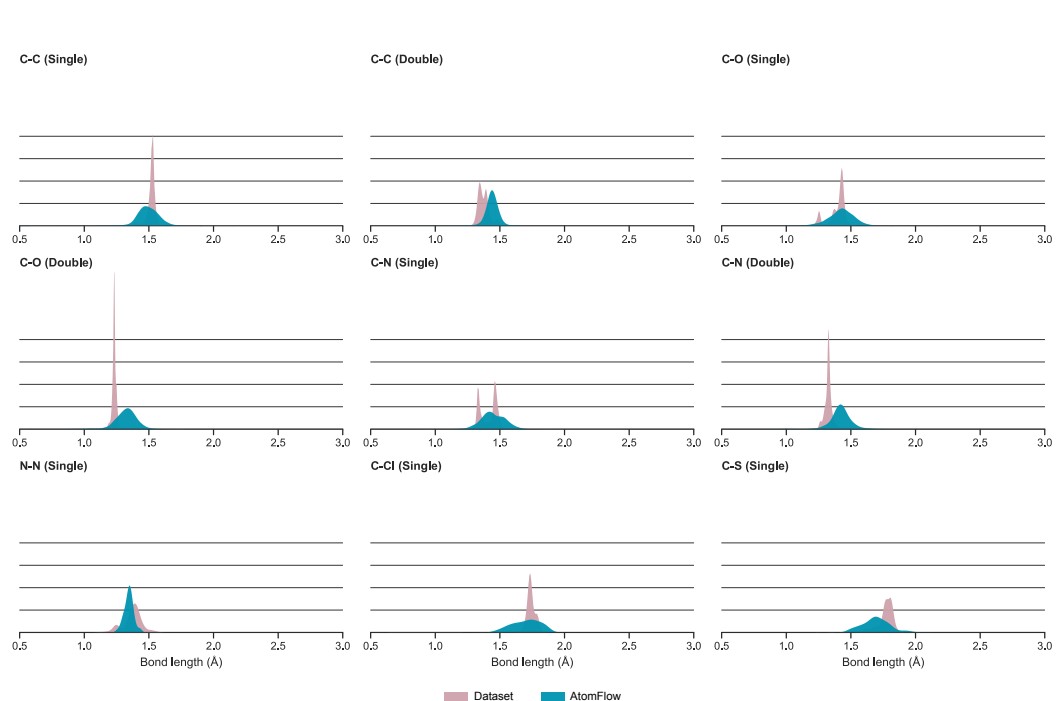

Figure S3: Chemical bond distribution of AtomFlow generated ligands for the extended set and ground truth ligands in the PDBBind dataset.

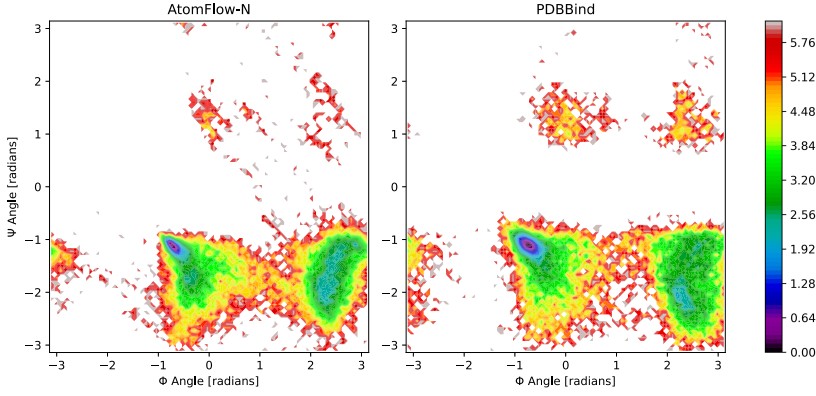

Figure S4: The Ramachandran plots for the generated protein (left) and the PDBBind protein (right), which demonstrate comparable **coverage** in the primary secondary structure regions.

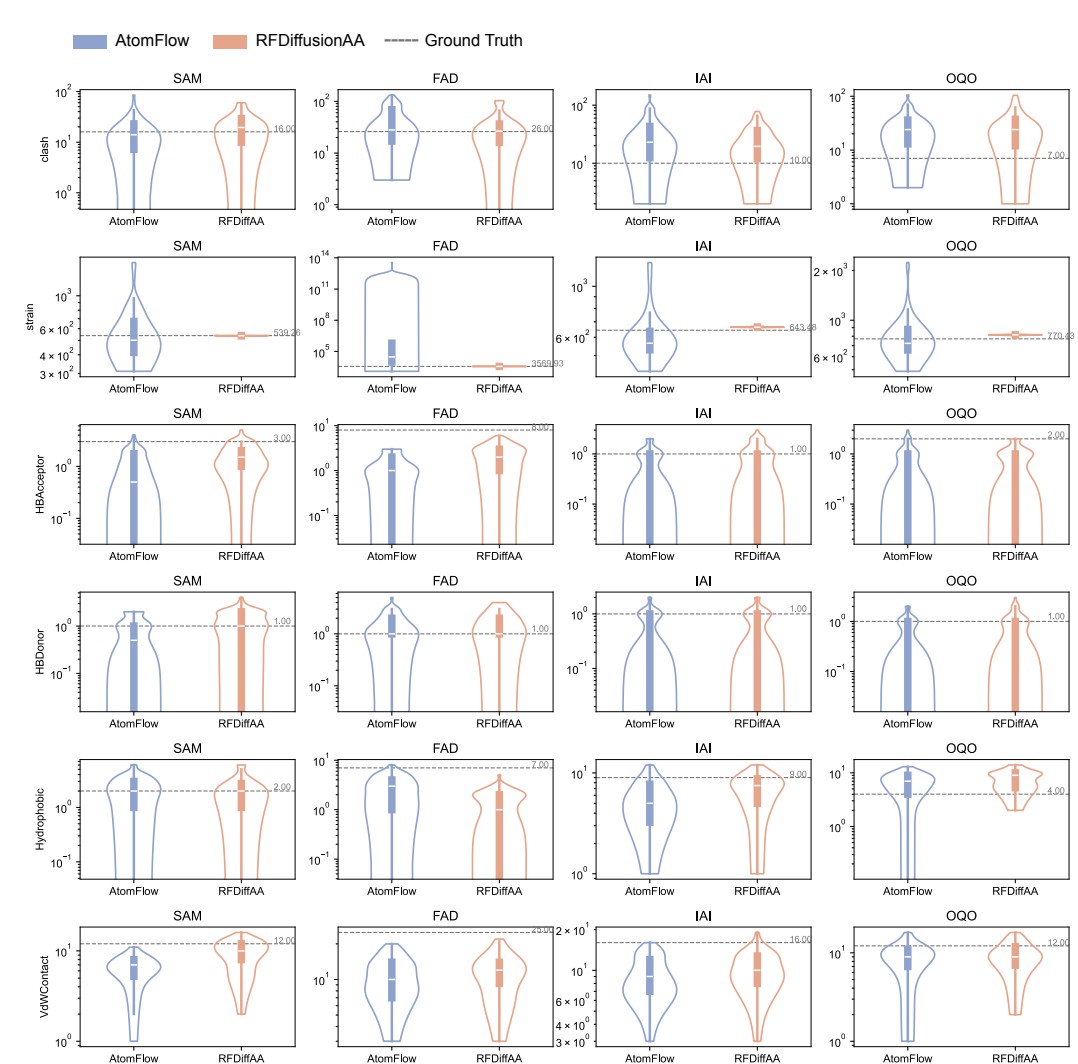

Figure S5: PoseCheck results of AtomFlow generated, RFDiffusionAA generated, and ground truth complex for the evaluation set.

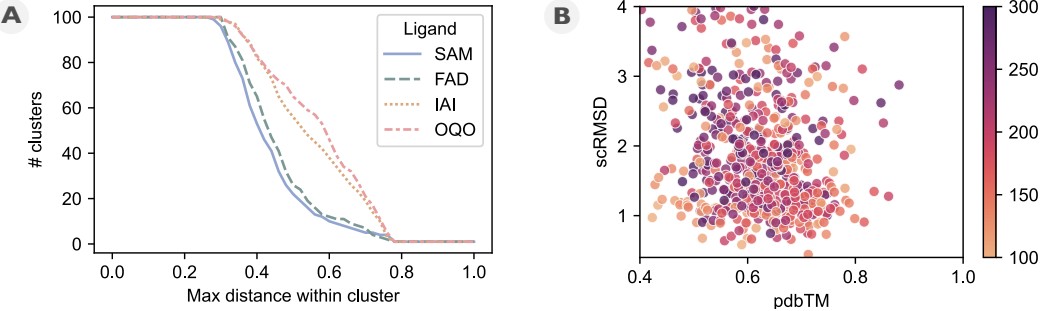

Figure S6: **A**: Cluster count based on different thresholds of the maximum difference within the cluster for each ligand in the evaluation set. **B**: Scatter plot of designability (scRMSD) vs. novelty (pdbTM) for ligands in the evaluation set.

