# OpenReview forum: "Design of Ligand-Binding Proteins with Atomic Flow Matching"
_ICLR.cc/2026/Conference — Submitted to ICLR 2026_

### Official Review · Reviewer_21E6 · 2025-10-22

**Soundness:** 3
**Presentation:** 2
**Contribution:** 2
**Rating:** 4
**Confidence:** 4

**Summary:**

This paper introduces AtomFlow, a flow-matching framework for protein-ligand complex generation condition on molecular ligand graphs. design. Protein and ligand representations are unified via "biotokens". The papers claims that the proposed model can perform de novo protein binder design without requiring a predefined ligand conformer, achieving comparable or superior binding affinity to RFDiffusionAA and faster inference speed.

**Strengths:**

- Biotoken representation: a unified feature framework for ligand atoms and protein residues within an SE(3)-equivariant space is elegant and facilitates joint modeling of multiple biomolecular types.

- Speedup: AtomFlow achieves over 5x inference speed compared to RFDiffusionAA.

**Weaknesses:**

1. The model predicts only C$-\alpha$ coordinates for proteins. Therefore, it is unclear to me how it can achieve high binding affinity when full protein information is not generated directly by the model but instead depends on downstream pipelines. It is also unclear whether, when using AutoDock Vina, the generated complexes were redocked or if the Vina score was simply computed on the generated structures. If the former is true, such discrepancies would weaken the claims regarding physical binding accuracy. Finally, some of the reported metrics, such as PoseBuster and PoseCheck, are mentioned as being performed but are not actually presented in the paper (see Appendix A.6, page 18). These metrics are essential for assessing the quality of the generated structures.

2. Equation 5 bins continuous distance values before applying flow matching, introducing a discontinuity that may conflict with the continuous-space assumptions of the flow ODE formulation.

3. The handling of categorical variables (e.g., residue types) is not discussed, even though flow matching operates in a continuous space.

4. The Vina score distributions for AtomFlow, reported in Equation 5, are quite similar to those of the competing method, RFDiffusionAA. These results, combined with the absence of key metrics such as PoseBuster and PoseCheck, weaken the overall contributions of the paper. However, I acknowledge that the reported speed-up remains a valid and valuable contribution.

5. The relationship of $(r_i, t_i)$ with respect to $a_i$  in the set  {$T_i = (r_i, t_i) \ | \ a_i \in \mathcal{P}$}  on page 3 line 135, is not specified. Also $t$ is used for both time and $t_{i,j}$ in Eq. 5.

**Questions:**

1. How are the categorical residue or atom types incorporated into the continuous flow-matching framework?

2. Can you provide explicit mathematical expressions for how $f^{pair}$ and $f^{token}$ are computed and used by the structure prediction network?

3. When using AutoDock Vina, are the generated complexes redocked or if is Vina simply computed on the generated structures?

4. Why is the PoseBusters/PoseCheck evaluation missing?

5. Could you discuss why continuous variables are discretized in Eq. (5), while still using a flow matching that handles continuos variables and whether this affects training stability?

---

> ### Author Response · Authors · 2025-11-18
>
> Thank you for your thoughtful comments! We address each point below.
>
> ### W1. Cα-only representation, Vina protocol, and missing pose-quality metrics
>
> **(a)** Although our model outputs only the backbone, the backbone geometry defines the binding pocket and largely shapes the spatial constraints that downstream sequence-design models rely on. A reasonable pocket can often guide the redesign process toward sequences with high binding affinity.
>
> **(b)** For AutoDock Vina, we evaluate scores directly on the generated complexes. Redocking is not allowed; only Vina’s small internal local optimization is used.
>
> **(c)** We apologize for the oversight. The PoseCheck caption in Figure S5 was written incorrectly, which may have made it hard to locate, and the PoseBusters results were unintentionally omitted during typesetting. The correct PoseBusters statistics are:
>
> - **AtomFlow-H:** 71.8% pass all 19 tests; 98.8% pass ≥18.
> - **AtomFlow-N:** 51.4% pass all 19 tests; 82.3% pass ≥18.
>
> We will correct this in the revision. RFDiffusion-AA requires an accurate ligand conformer, so a fair comparison is not available for cases without reliable conformers; therefore we did not include PoseBusters comparisons against that baseline. However, for PoseCheck we do provide the baseline results and ground-truth reference values.
>
> ### W2. Discretization in Eq. (5)
>
> The binned-distance representation follows the AlphaFold2 template module. In early experiments, continuous coordinates did not offer clear gains, while binned distances captured structural information effectively and remained stable. More expressive continuous modules are a promising future direction.
>
> ### W3. Handling categorical residue types
>
> We focus on generating plausible backbones rather than residue identities. This matches practical workflows [1-3] where many backbones are generated first and sequences are redesigned afterward. Current all-atom generative approaches still underperform MPNN-style redesign in sequence quality when it comes to designability and developability, so we intentionally avoid direct categorical generation here.
>
> ### W4. Comparison to RFDiffusion-AA
>
> AtomFlow does not require a ligand conformer. For ligands without reliable structural data, force-based conformer generation introduces systematic error that directly harms RFDiffusion-AA. In contrast, AtomFlow-N infers the pose jointly with the protein and remains stable in these low-information settings. For well-characterized ligands, providing desirable crystal poses to RFDiffusion-AA is indeed effective.
>
> ### W5. Notation in $\{T_i = (r_i, t_i) \mid a_i \in \mathcal{P}\}$
>
> Our intention was simply that $i$ indexes valid elements of $\mathcal{P}$, and the notation will be revised for clarity. The repeated use of $t$ will be replaced.
>
> ### Q1. Incorporation of categorical residue/atom types
>
> As noted in W3, we do not generate residue identities. Existing strategies include separate flows for categorical variables [4] or atom14-based generation [5-7], but neither produced sequence quality comparable to redesign. Backbone generation remains the more reliable choice.
>
> ### Q2. Explicit forms of $f^{pair}$ and $f^{token}$
>
> All features are described in Appendix A.2. Each feature is one-hot encoded and linearly projected before forming the pair and token representations. We will expand this section with explicit formulations.
>
> ### Q3. Vina evaluation
>
> Same as W1(b): we compute Vina scores on the generated structures without redocking.
>
> ### Q4. PoseBusters/PoseCheck
>
> Answered in W1(c). Full results will be added to revision.
>
> ### Q5. Discretization vs. continuous flow
>
> See W2. The discretization follows established practice from AlphaFold2 and did not harm stability or performance in our setting.
>
> Thank you again for your time and constructive feedback. We look forward to your response!
>
>
> [1] Krishna, R., Wang, J., ... & Baker, D. (2024). Generalized biomolecular modeling and design with RoseTTAFold All-Atom. *Science*, *384*(6693), eadl2528.
>
> [2] Bennett, N. R., Watson, J. L. ... & Baker, D. (2025). Atomically accurate de novo design of antibodies with RFdiffusion. *bioRxiv*, 2024-03.
>
> [3] Zambaldi, V., La, D., ... & Wang, J. (2024). De novo design of high-affinity protein binders with AlphaProteo. arXiv preprint arXiv:2409.08022.
>
> [4] Campbell, A., Yim, J., ... & Jaakkola, T. (2024). Generative flows on discrete state-spaces: Enabling multimodal flows with applications to protein co-design. *arXiv preprint arXiv:2402.04997*.
>
> [5] Qu, W., Guan, J., Ma, R., Zhai, K., Wu, W., & Wang, H. (2024). P (all-atom) is unlocking new path for protein design. *bioRxiv*, 2024-08.
>
> [6] BoltzGen: Toward Universal Binder Design
>
> [7] Corley, N., Mathis, S., Krishna, R., Bauer, M. S., Thompson, T. R., Ahern, W., ... & DiMaio, F. (2025). Accelerating biomolecular modeling with atomworks and rf3. *bioRxiv*, 2025-08.

---

> > ### Comment · Reviewer_21E6 · 2025-11-28
> > **Response**
> >
> > Thanks for your answers but I still have some concerns
> > 1. Thanks for adding the PoseBuster evaluation, but since your comparing against RFDiffusion-AA, it feels a bit odd to me that you don't compare PoseBuster against it. Also, what do you mean by an accurate ligand conformer?
> >
> > 2. I am still a bit puzzled about data representation and flow matching model. Are you using discrete data on a flow matching model that handle continuos variables?

---

> > > ### Author Response · Authors · 2025-11-28
> > >
> > > Thank you for the follow-up. We clarify the two points below.
> > >
> > > **1. About PoseBusters and the comparison with RFDiffusion-AA**
> > >
> > > We use PoseBusters to evaluate the intrinsic physical quality of the ligand conformer itself, and use PoseCheck to evaluate the binding interface. RFDiffusion-AA takes a fixed ligand conformer as input and outputs it without modification, so its score is entirely determined by the input and does not reflect model behavior. AtomFlow, in contrast, must construct the bound ligand conformer, so PoseBusters is informative for AtomFlow but not suitable for comparison with RFDiffusion-AA.
> > >
> > > **2. About flow matching and the representation**
> > >
> > > Our flow matching formulation is defined directly on continuous 3D coordinates. The only implementation detail that may cause confusion is that the denoising network receives a binned coordinate representation rather than exact atomic positions. This is a pragmatic choice to improve stability for AF-2–style architectures, since these models prefer pairwise distance features and the binned representation provides a smoother and more robust input than raw continuous distances. This does not alter the fact that the underlying variables and the flow dynamics are fully continuous.
> > >
> > > We hope this resolves the concerns, and we are happy to provide further details if needed!

---

### Official Review · Reviewer_JiYA · 2025-10-31

**Soundness:** 4
**Presentation:** 4
**Contribution:** 3
**Rating:** 6
**Confidence:** 3

**Summary:**

This paper addresses an important challenge in molecular AI: designing protein binders for small-molecule ligands. The proposed method, **ATOMFLOW**, generates binding coordinates for protein-ligand complexes using only the 2D molecular graph of the target ligand. This is achieved through biotokenization of ligand atoms and protein residues, pairwise relationship embedding, and a flow-matching framework. The method is supported by thorough experiments across diverse ligands.

However, the novelty of the work is limited. The problem formulation and methodological components—such as flow matching and SE(3)-equivariant modeling—are largely adapted from prior literature. While the results are competitive, they do not demonstrate significant improvements over existing models such as RFDiffusionAA or Chai-1.

**Strengths:**

- Clear problem formulation and modular architecture.
- Robust evaluation across ligands.
- Faster inference compared to RFDiffusionAA.
- Unified representation of protein and ligand tokens.

**Weaknesses:**

- Limited methodological novelty.
- Benchmarking setup may introduce bias due to self-generated reference structures.

**Questions:**

1. **Terminology clarity**: Please expand acronyms such as ODE and SAM. Given the diversity of the AI4Science community, not all readers share the same technical background.

2. **Line 205**: What does the symbol \( Q \) represent? Clarifying this would improve readability.

3. **One-hot binned distance map**: What motivated this design choice? How does it compare to alternatives such as continuous embeddings or radial basis functions?

4. **Choice of LigandMPNN and ESMFold**: Why were these models selected for sequence recovery and structure validation? A brief overview of their accuracy and relevant references would be helpful.

5. **Benchmark fairness (Table 2)**: The reference structures are derived from your own pipeline using LigandMPNN and ESMFold. Does this introduce bias when comparing against other models? Please clarify.

6. **Feature redundancy**: Figure 2 shows that the distance map is passed to the Feature Embedder. If pairwise features already encode distance, is there redundancy between `pair feat` and the distance map? Clarifying their distinct roles would strengthen the architectural rationale.

---

> ### Author Response · Authors · 2025-11-18
>
> Thank you for your thoughtful comments! We address each point below.
>
> ### 1. Terminology clarity
>
> We will expand all acronyms upon first use. ODE refers to *ordinary differential equation*. SAM**,** FAD, IAI, OQO, and similar terms are ligand codes from the Chemical Component Dictionary. These clarifications, as well as the definition for all acronyms will be added.
>
> ### 2. Meaning of Q
>
> The symbol Q refers to the quotient space mentioned in the previous sentence. We will define Q explicitly at its first appearance.
>
> ### 3. One-hot binned distance map
>
> This design was inspired by the template module in AlphaFold2. In our preliminary experiments, binned distance representations were already sufficient for the model to capture structural information, and directly supplying precise coordinates did not provide noticeable improvement. We consider more sophisticated coordinate-aware modules a promising direction to improve performance.
>
> ### 4. Choice of LigandMPNN and ESMFold
>
> LigandMPNN is a backbone-conditioned sequence design model with similar performance to ProteinMPNN on sequence recovery and structural consistency, and improved handling of protein–ligand contacts. This backbone→sequence step is widely used and experimentally validated.
>
> ESMFold is a fast structure predictor built on the ESM protein language model and is commonly adopted as a lightweight alternative to AlphaFold with a small accuracy gap on stable *de novo* structures. We will briefly introduce both models in the revision.
>
> ### 5. Benchmark fairness
>
> The pipeline follows the standard practice in current *de novo* design evaluations [1-3], though we agree that using LigandMPNN + ESMFold to generate reference structures may introduce bias. The major uncertainty comes from LigandMPNN potentially failing to design the optimal sequence for certain backbones. To reduce this, we selected the best sequence among eight designs. We will clarify this in the paper.
>
> This setup may slightly favor RFDiffusionAA, as LigandMPNN and RFDiffusionAA are both developed in the Baker lab, and their models share data processing conventions. Furthermore, the Baker lab recently open-sourced the AtomWorks ****[4] framework, which unifies data handling across their models.
>
> ### 6. Feature redundancy
>
> The residue-distance feature for proteins in Figure 2 is included to illustrate the framework’s extensibility, particularly when known structural motifs require additional pairwise constraints. We did not conduct experiments using these extra features; they are shown only to demonstrate compatibility. We will revise the text to make this clearer.
>
> ### 7. Methodology novelty
>
> We agree that our method shares some structural similarities with prior flow-based models. Prior de novo design methods typically relied on non–SE(3)-invariant losses. To our knowledge, we are the first to bring an AlphaFlow-style SE(3)-invariant loss into a de novo ligand-binding setting and show how to make it work for both protein residues and ligand atoms.
>
> A second contribution is the evaluation itself. Building on the RFDiffusion-AA benchmarks, we added a much more complete set of structural and physicochemical analyses, and we introduced iPAE as an affinity-related metric. This helps form a more comprehensive and practical evaluation suite for de novo ligand-binding design.
>
> Thank you again for your time and constructive feedback. We look forward to your response!
>
> [1] Krishna, R., Wang, J., Ahern, W., Sturmfels, P., Venkatesh, P., Kalvet, I., ... & Baker, D. (2024). Generalized biomolecular modeling and design with RoseTTAFold All-Atom. *Science*, *384*(6693), eadl2528.
>
> [2] Bennett, N. R., Watson, J. L., Ragotte, R. J., Borst, A. J., See, D. L., Weidle, C., ... & Baker, D. (2025). Atomically accurate de novo design of antibodies with RFdiffusion. *bioRxiv*, 2024-03.
>
> [3] Zambaldi, V., La, D., Chu, A. E., Patani, H., Danson, A. E., Kwan, T. O., ... & Wang, J. (2024). De novo design of high-affinity protein binders with AlphaProteo. arXiv preprint arXiv:2409.08022.
>
> [4] Corley, N., Mathis, S., Krishna, R., Bauer, M. S., Thompson, T. R., Ahern, W., ... & DiMaio, F. (2025). Accelerating biomolecular modeling with atomworks and rf3. *bioRxiv*, 2025-08.

---

> > ### Comment · Reviewer_JiYA · 2025-11-21
> >
> > Thank you for the responses, and they addressed most of my questions. I loook forward to seeing the updated Figure 2 once it is publically online. The responses enhanced my confidence in my ratings, and I will keep them unchanged. Thanks!

---

> > > ### Author Response · Authors · 2025-11-25
> > >
> > > We are very glad to hear that our responses helped clarify your question! We have now uploaded the updated version of the paper, including the new Figure 2 and its caption. Thank you again for your valuable feedback and for your recognition of our work.

---

### Official Review · Reviewer_nHfv · 2025-11-02

**Soundness:** 3
**Presentation:** 3
**Contribution:** 2
**Rating:** 4
**Confidence:** 4

**Summary:**

This paper presents AtomFlow, a flow matching model for designing a protein structure to bind a small molecule ligand. The model jointly denoises the structure of the protein and ligand and thus does not require knowledge of the ligand pose, unlike RFDiffusionAA. The architecture is based on AlphaFold and predicts denoised structures from a distance map input of the input structure. The method is evaluated on (1) the set of 4 ligands studied in RFDiff-AA, and (2) an expanded set of ligands curated by the authors. AtomFlow is shown to have comparable or better designability than RFAA as well as similar Vina score.

**Strengths:**

* The work is solidly executed, with sensible architectural choices, strong initial evaluations, and clear and concise writing.
* The task tackled is significant and the competitive results signify well-executed model engineering and training practices.
* The authors re-derive the quotient-space flow matching fromework from AlphaFlow with more solid theoretical justification.
* The paper is quite clearly written. The figures are well made, visually appealing, and informative.

**Weaknesses:**

**Originality**
* The methodology can be described as a flow-matching version of RFDiff-AA and does not score high on originality / novelty from a ML perspective. Further, the flow model architecture and noising process are based on AlphaFlow, with different justification but no difference in practice as far as I can tell. To improve on this axis, while it's not clear that more methodological novelty is needed for its own sake, the authors could focus on novel evaluations or applications of the proposed method.

**Quality**
* The computational evaluations are well executed, but limited in scope. Most of the analysis focuses on only 4 ligands, raising concerns about sample size and statistical significance.
* The diversity and novelty evaluations are nice, but only AtomFlow is evaluated, not RFDiff-AA or the other baselines.

**Significance**

* The overall significance of the contribution is unclear as it represents an incremental methodological advance over RFDiff-AA with more or less the same model capabilities. The authors argue that not needing to specify the ligand pose is a big plus, but no meaningful evidence or use case is provided for this distinction. After all, RFDiff-AA has been experimentally validated, whereas AtomFlow-generated poses have not. It is of course not expected for a ML submission to experimentally validate the proteins, but it should be made a bit clearer why the main point of difference with RFDiff-AA is important to tackle as a ML problem.

**Questions:**

No specific questions.

---

> ### Author Response · Authors · 2025-11-18
>
> Thank you for your thoughtful comments! We address each point below.
>
> ### 1. On originality
>
> We agree that our method shares some structural similarities with prior flow-based models. What we want to highlight is that AlphaFlow is essentially an alternative conformer generation model, adapted from finetuning AlphaFold. In other words, AlphaFlow is still a *predictive* model based on an existing structure context.
>
> In contrast, our work focuses on *de novo* protein generation, where the model must construct the full complex from scratch. Prior de novo design methods typically relied on non–SE(3)-invariant losses [1-3]. To our knowledge, we are the first to bring an AlphaFlow-style SE(3)-invariant loss into a de novo ligand-binding setting and show how to make it work for both protein residues and ligand atoms.
>
> A second contribution is the evaluation itself. Building on the RFDiffusion-AA benchmarks, we added a much more complete set of structural and physicochemical analyses, and we introduced **iPAE** as an affinity-related metric. This helps form a more comprehensive and practical evaluation suite for de novo ligand-binding design.
>
> ### 2. On the scope of evaluations
>
> We are fully aware that a small set of ligand does not reflect the whole picture of model performance. We may not have made it sufficiently clear in the main text, but we have already included an **extended evaluation on 20 ligands** in the Appendix. The trends remain consistent with those in the main text.
>
> For novelty and diversity, we have provided RFDiffusion-AA’s results in the Appendix on the last page as well, and we apologize for not highlighting in the main text that these results are provided in the Appendix. As mentioned in main text, AtomFlow underperforms on these metrics. The main limitation here is the difference in training data scale. Prior studies have shown that dataset size strongly governs these metrics [2]. We plan to address this through larger-scale training.
>
> ### 3. On significance
>
> In the paper, we include a case study showing that for less-studied ligands, where no reliable conformer is available, the generation quality of RFDiffusion-AA drops noticeably. In contrast, AtomFlow-N, which does not rely on a predefined pose and instead infers the ligand pose jointly with the protein, produces much more stable results.
>
> In practical workflows, when a ligand has been well characterized and high-quality crystal structures exist, providing accurate poses from the PDB to RFDiffusion-AA is a standard and effective practice. The difficulty appears when the ligand lacks sufficient structural data. In that case, users must generate artificial poses using force-based methods, and these approximated conformers often introduce systematic errors that directly degrade RFDiffusion-AA’s performance. Since the correctness of the input conformer strongly influences the output quality, developing models that can generate the full conformer directly from the ligand SMILES becomes highly valuable for real applications.
>
> Further more, AtomFlow generates 5x faster than RFDiffusionAA, which reduces the computational burden when using RFDiffusionAA in the lab, and this improvement was also appreciated by reviewer 21e6.
>
> We also acknowledge that AtomFlow has not yet undergone experimental validation, while RFDiffusion-AA has. Our focus here is to provide a principled modeling framework for de novo generation and a more complete *in silico* computational evaluation. Experimental validation is an important next step.
>
> Thank you again for your time and constructive feedback. We look forward to your response!
>
>
> [1] Bennett, N. R., Watson, J. L., Ragotte, R. J., Borst, A. J., See, D. L., Weidle, C., ... & Baker, D. (2025). Atomically accurate de novo design of antibodies with RFdiffusion. *bioRxiv*, 2024-03.
>
> [2] Huguet, G., Vuckovic, J., Fatras, K., Thibodeau-Laufer, E., Lemos, P., Islam, R., ... & SequenceAugmented, S. E. Flow Matching For Conditional Protein Backbone Generation, May 2024.
>
> [3] Geffner, T., Didi, K., Zhang, Z., Reidenbach, D., Cao, Z., Yim, J., ... & Kreis, K. (2025). Proteina: Scaling flow-based protein structure generative models. *arXiv preprint arXiv:2503.00710*.

---

### Official Review · Reviewer_3bnS · 2025-11-03

**Soundness:** 3
**Presentation:** 3
**Contribution:** 2
**Rating:** 4
**Confidence:** 4

**Summary:**

The paper proposes ATOMFLOW, which uses a unified biotoken representation to jointly generate protein and ligand structures
by learning the distribution of token positions conditioned on a ligand chemical graph. It uses flow matching to learn the structure prediction model based on both token features and pair features. The proposed method is evaluated on the PDBBind dataset, with a focus on four ligands (FAD, SAM, IAI, OQO).

**Strengths:**

The paper targets at an important and valuable task and achieves comparable performance to RFDiffusionAA.

**Weaknesses:**

The weaknesses of this paper are listed as follows:

1. The evaluation of the method just focuses on several ligands, making the generalization of the method unknown. Basically, most of the experiments are conducted on four specific ligands. I'm not sure if the designed methods can be well adapted to broader applications, like enzyme-substrate complex structure prediction, which is even harder as the transition state of the enzyme is generally hard to capture.

2. It seems the proposed method is limited to generate shorter proteins. All the designed proteins are limited to a length shorter than 300.

3. The performance of the original ATOMFlOW-N is worse than RFdiffusionAA, and the variant ATOMFlOW-H is comparable to RFDiffusionAA.

4. There are some newer models like RFDiffusion2, which the proposed method should be compared with.

5. In Figure 8, when showing the novelty and diversity of the proposed method, the paper doesn't provide the performance of baseline methods like RFDiffusionAA, which makes it hard to capture the comparison with previous SOTA methods.

**Questions:**

The ATOMFLOW-N performs worse than RFDiffusionAA, while ATOMFLOW-H which uses an auxiliary hint input of the pairwise distance matrix of the bound structure achieves better performance. Can the authors explain in detail how they implemented ATOMFLOW-H and why it helped?

---

> ### Author Response · Authors · 2025-11-18
>
> Thank you for your thoughtful comments! We address each point below.
>
> ### 1. On generalization and the applicability to enzyme design
>
> We are fully aware that a small set of ligand does not reflect the whole picture of model performance. We may not have made it sufficiently clear in the main text, but we have already included an extended evaluation on 20 ligands in the Appendix. The trends remain consistent with those in the main text.
>
> As you noted, enzyme design is considerably more challenging, since it requires detailed understanding of intermediate or transition-state geometries, and such data are still limited. In our own pipeline, RFDiffusion-AA also rarely succeeds on de novo enzyme design.
>
> At present, a practical approach in the community is to first understand the mechanism, fix the spatial arrangement of a few key residues that interact with the small molecule, and then carry out large-scale design and screening. RFDiffusion2 offers a convenient interface for fixing multiple noncontiguous residues, and we are actively studying this direction. Our current model does not yet support this functionality.
>
> ### 2. On the length of designed proteins
>
> In de novo binder design, shorter proteins are generally preferred. The typical workflow includes backbone design, sequence design, screening, and experimental validation [1–3]. Larger proteins are more difficult to synthesize and validate experimentally, and many desired functions can be achieved with sequences of about 50–200 amino acids. Based on these practical considerations, we did not emphasize performance on very large proteins in the paper.
>
> ### 3. On the performance gap between AtomFlow-N and AtomFlow-H
>
> AtomFlow-N is designed for the setting where the ligand pose is not provided, so the model must infer both the binding pose and the protein structure, which is a more challenging problem. AtomFlow-H, in contrast, receives an auxiliary hint derived from the ground-truth binding pose: a binned pairwise distance matrix that is first one-hot encoded and then projected through a linear layer before being provided to the pairformer stack. This naturally improves performance, as the model is effectively given a high-quality conformer pose extracted directly from the ground truth.
>
> Since RFDiffusion-AA requires a known ligand pose as input, we introduced AtomFlow-H to allow for a fair comparison under matching conditions. In our own research pipeline, we use AtomFlow-N primarily because it enables us to tackle this extended and harder problem where the ligand pose is unknown.
>
> ### 4. On comparison with RFDiffusion2
>
> RFDiffusion2 is mainly designed for situations involving fragmented or noncontiguous motifs, especially in enzyme design where several spatially adjacent but sequence-distant residues must be fixed. Our study focuses on motif-free de novo design, which is a different problem setting. Therefore, we did not include RFDiffusion2 as a primary baseline. We have clarified this distinction in the related work section in the updated version to avoid confusion.
>
> ### 5. On novelty and diversity comparisons
>
> The Appendix includes the corresponding RFDiffusion-AA results on its final page, and we apologize that this was not explicitly highlighted in the main text. As discussed in the main text, the difference in training data scale leads to a noticeable gap in novelty and diversity. Prior studies have shown that dataset size strongly affects these metrics [4], so this remains a known limitation. We plan to scale up training to further improve performance.
>
> Thank you again for your time and constructive feedback. We look forward to your response!
>
> [1] Krishna, R., Wang, J., Ahern, W., Sturmfels, P., Venkatesh, P., Kalvet, I., ... & Baker, D. (2024). Generalized biomolecular modeling and design with RoseTTAFold All-Atom. *Science*, *384*(6693), eadl2528.
>
> [2] Bennett, N. R., Watson, J. L., Ragotte, R. J., Borst, A. J., See, D. L., Weidle, C., ... & Baker, D. (2025). Atomically accurate de novo design of antibodies with RFdiffusion. *bioRxiv*, 2024-03.
>
> [3] Zambaldi, V., La, D., Chu, A. E., Patani, H., Danson, A. E., Kwan, T. O., ... & Wang, J. (2024). De novo design of high-affinity protein binders with AlphaProteo. arXiv preprint arXiv:2409.08022.
>
> [4] Huguet, G., Vuckovic, J., Fatras, K., Thibodeau-Laufer, E., Lemos, P., Islam, R., ... & SequenceAugmented, S. E. Flow Matching For Conditional Protein Backbone Generation, May 2024.

---

### Official Review · Reviewer_5EfC · 2025-11-11

**Soundness:** 4
**Presentation:** 4
**Contribution:** 4
**Rating:** 8
**Confidence:** 4

**Summary:**

This paper proposes a method for protein binder design. Both protein residues and ligand are modeled in a joint representation ("biotokens"), which enables to jointly generate protein-ligand complexes in an SE(3)-equivariant flow matching framework.

**Strengths:**

The paper is well written and the presentation is clear. The approach of representing both ligand and residues in a joint feature representation is elegant and enables to use the same flow matching generative process for both, leading to a generative model for protein-ligand complexes. The method is on par and sometimes outperforms RFDiffusionAA in binder design quality, and (through flow matching) offers faster inference.

**Weaknesses:**

No major weaknesses. Great paper!

Technical remarks:

Typo in Fig2: Piror Distribution

**Questions:**

'

---

> ### Author Response · Authors · 2025-11-18
>
> Thank you sincerely for the kind and encouraging review. We are very grateful for the attention you gave the work, and your positive assessment really means a lot to us.
> We have also noted the small typo you mentioned and have corrected it in the updated version.
> Thank you again for your time, your confidence, and the thoughtful feedback. We will keep refining the paper to present it in the best shape possible. (•̀ᴗ•́)و

---

### Author Response · Authors · 2025-12-01
**Final Remarks by Authors**

We sincerely thank all reviewers for their thoughtful feedback. We are encouraged that most reviewers recognized the quality of our execution, the clarity of the writing, and the significance of the problem we aim to solve. In our individual responses, we addressed every comment in detail and revised the manuscript accordingly. All revisions mentioned in our responses are already incorporated into the current manuscript and highlighted in *blue*.

The main concerns about the completeness of our evaluation were largely due to insufficient guidance to the relevant appendix in the main text, which led some reviewers to overlook the results provided there. For example, questions regarding the breadth of our evaluation and the absence of baseline (reviewer 3bnS, nHfv) stemmed from the fact that we did not clearly direct readers to the extra evaluation set of 20 ligands and the additional baseline results reported in the appendix, which further support the conclusions presented in the main text. We have now clarified these connections in the revised manuscript and also fixed several minor clarity issues (reviewer JiYA, 21E6) highlighted by the reviewers.

Regarding concerns about the novelty of our work (nHfv, JiYA), we have provided detailed clarifications in the corresponding rebuttals. While perspectives on *novelty* may naturally vary, we carefully explained how our approach differs from existing methods beyond the novel practical setting of designing binders from ligand molecule graph instead of a predefined conformer, including the algorithmic advances for de novo protein generation, the introduction of a novel and comprehensive evaluation suite, and improved computational efficiency.

We believe the revised version now presents the methodological details and experiment results in a more accessible manner. We once again thank all reviewers for their constructive input, and we hope that this summary of our rebuttal helps the Area Chair in making an informed decision.

---

### Meta-Review · Area_Chair_KESj · 2026-01-06

**Summary:**

In this submission, the authors propose an atomic FM method to generate/design proteins given a small molecule (i.e., ligand). The proposed method does not require knowledge of binding poses and generates proteins using an SE(3)- equivariant structure prediction network. Experiments show the proposed method's potential to some degree.

In my opinion, this submission reflects a common problem in the AI4Science direction of machine learning papers: If the main motivation and contribution of a paper lies in proposing new theories or technologies and then using problems in the Science domain to achieve verification, then the novelty of the method and the insights it provides are crucial. If the purpose of the paper is to solve a specific problem in the Science domain, then performance, especially its advantage over cutting-edge technologies in that field, becomes the most important evaluation metric. From this perspective, the method proposed in this article has not been compared with RFDiffusion2 in terms of performance, and its novelty is somewhat lacking.

**Reviewer Concerns:**

The main concerns of reviewers include 1) the generalizability of the method in desiging longer proteins for various ligands, 2) the novelty of the method itself, and 3) the lack of comparison with strong baselines like RFDiffusion2. Although the authors provide more explanations, I think the concerns are not resolved.

**Reviewer Scores:**

I think Reviewer 5EfC would have reduced his/her score to 6 or 4 if he/she had discussed with other reviewers.

---

### Decision · Program_Chairs · 2026-01-26

Reject